# AG-REPA: Causal Layer Selection for Representation Alignment in Audio Flow Matching

Pengfei Zhang [1]  Tianxin Xie [1]  Minghao Yang [1]  Li Liu [1] [*]

## Abstract

REPresentation Alignment (REPA) improves the training of generative flow models by aligning intermediate hidden states with pretrained teacher features, but its effectiveness in token-conditioned audio Flow Matching critically depends on the choice of supervised layers, which is typically made heuristically based on the depth. In this work, we introduce **A**ttribution-**G**uided **REP**resentation **A**lignment (**AG-REPA**), a novel causal layer selection strategy for representation alignment in audio Flow Matching. Firstly, we find that layers that best store semantic/acoustic information (high teacher-space similarity) are not necessarily the layers that contribute most to the velocity field that drives generation, and we call it **S**tore-**C**ontribute **D**issociation (**SCD**). To turn this insight into an actionable training guidance, we propose a forward-only gate ablation (FoG-A) that quantifies each layer's causal contribution via the induced change in the predicted velocity field, enabling sparse layer selection and adaptive weighting for alignment. Across unified speech and general-audio training (LibriSpeech + AudioSet) under different token-conditioning topologies, AG-REPA consistently outperforms REPA baselines. Overall, our results show that alignment is most effective when applied to the causally dominant layers that drive the velocity field, rather than to layers that are representationally rich but functionally passive.

**Code Availability.** The official code repository is available at https://github.com/zpforlove/AG-REPA.

---

[*]Corresponding author.  [1]AI Thrust, Information Hub, The Hong Kong University of Science and Technology (Guangzhou). Correspondence to: Li Liu <avrillliu@hkust-gz.edu.cn>.

*Proceedings of the 43$^{rd}$ International Conference on Machine Learning*, Seoul, South Korea. PMLR 306, 2026. Copyright 2026 by the author(s).

## 1. Introduction

Flow Matching (FM) models (Lipman et al., 2023) have emerged as a dominant paradigm in audio generation, achieving strong performance in speech synthesis (Le et al., 2023; Chen et al., 2025; Du et al., 2024; Mehta et al., 2024) and general audio synthesis (Vyas et al., 2023; Guan et al., 2024). These models learn a continuous velocity field that transports samples along efficient trajectories from a simple prior to the target data distribution. While effective, training these models is computationally expensive. REPresentation Alignment (REPA) (Yu et al., 2025; Wang et al., 2025; Singh et al., 2026) has emerged as a promising acceleration technique by supervising intermediate layers with pretrained teacher features. However, existing alignment strategies encounter a significant methodological limitation: they depend predominantly on heuristic layer selection (fixed mid-layer alignment) or rely on cross-modal supervision (as seen in video-to-audio generation). This overlooks a fundamental question in the generative mechanism: *Does the layer that stores the most semantic information actually contribute the most to the audio generation process?*

Existing REPA-based methods (Wang et al., 2025; Singh et al., 2026) were primarily developed for vision tasks, where spatial structures in hidden states align naturally with visual encoders. Extensions to the audio domain (Ton et al., 2025; Shan et al., 2025) largely depend on video conditioning, utilizing dense visual features as anchors. This leaves token-conditioned audio synthesis unexplored, which poses a unique problem formulation: the model must decode sparse, discrete tokens into continuous waveforms without the aid of explicit visual grounding. If we simply follow prior practice and align at fixed mid-layers, we risk optimizing layers that store rich information but contribute little to the actual generation process.

In this work, we focus on a unified audio generation framework that performs both Text-to-Speech (TTS) and Text-to-Audio (TTA) synthesis within a single Flow Matching model. Unlike prior systems that train separate models for each domain, our architecture shares the same DiT-based FM backbone across speech and general audio generation, differing only in the upstream tokenization pathway (semantic tokens for speech, event-conditioned tokens for audio).

This unified paradigm offers an ideal setting for studying representation alignment: if intermediate-layer supervision can accelerate training, *where* should such alignment be applied when a single model must simultaneously master the distinct acoustic manifolds of human speech and environmental sounds? Addressing this question requires moving beyond simple fixed-layer strategies toward a principled understanding of how different layers *functionally* contribute to the shared velocity field across generation tasks.

Motivated by the distinction between representational and functional measures in neural network analysis (Klabunde et al., 2025), we conduct a systematic layer-wise analysis of token-conditioned FM models for audio generation. Our investigation reveals a counter-intuitive phenomenon we term **Store-Contribute Dissociation (SCD)** (Figure 1): while deep layers serve as the primary *storage* for rich semantics, it is the shallow layers that make the active *contribution* to the velocity field gradients driving the generation dynamics. This dissociation explains why heuristic alignment fails to maximize efficiency, as it often targets layers that "know" a lot but "do" little for the current velocity estimation.

Leveraging this insight, we propose **Attribution-Guided REPA (AG-REPA)**, a principled framework that shifts alignment from heuristic selection to functional targeting. By aligning only the *functionally critical layers* identified by FoG-A, AG-REPA ensures that supervision is applied where it impacts generation most.

In summary, the contributions of this work are threefold:

**1)** Firstly, through layer-wise quantitative analysis, we uncover a critical mismatch in token-conditioned audio generation: layers that serve as "semantic reservoirs" (high LASP alignment) are distinct from the *functionally critical* layers that drive the velocity field (high FoG-A attribution). This finding theoretically explains the inefficiency of heuristic, depth-based REPA alignment strategies.
**2)** Leveraging these findings, we then propose a causality-driven training strategy called AG-REPA that dynamically selects and weights layers based on their functional attribution rather than static heuristics.
**3)** Experiments on unified speech (LibriSpeech) and general audio (AudioSet) generation demonstrate that AG-REPA reduces Fréchet Audio Distance (FAD) by 18% and 16% respectively compared to the best single-fixed-layer REPA baseline.

**Conflict of Interest Disclosure**

The authors affirm that this work was conducted solely as independent academic research. No author maintains a financial relationship with, holds employment at, or possesses an equity interest in any organization whose products, models, or services are evaluated herein. To the best of our knowledge, no financial arrangements or affiliations exist that could reasonably be perceived as bearing upon the research design, experimental evaluation, interpretation of findings, or conclusions presented in this paper.

## 2. Related Work

### 2.1. Flow Matching for Audio Generation

Flow Matching (Lipman et al., 2023) has rapidly replaced standard diffusion models in audio generation due to its simpler training objective and efficient inference paths. Voicebox (Le et al., 2023) pioneered large-scale Flow Matching for speech, while Matcha-TTS (Mehta et al., 2024) demonstrated that Flow Matching with Optimal Transport yields high output quality with faster synthesis in fewer sampling steps. Building on these, recent works have scaled the paradigm to general audio (Vyas et al., 2023) and zero-shot TTS (Du et al., 2024; Chen et al., 2025). However, these works primarily focus on architecture (e.g., DiT vs. UNet) or data scaling, leaving the internal training behavior of such models largely underexplored. Our work complements this landscape by focusing on *training dynamics*, specifically dissecting layer-wise contributions to improve training efficiency via alignment.

### 2.2. Representation Alignment

Originally proposed for image generation, REPA (Yu et al., 2025) aligns DiT hidden states with self-supervised visual features (e.g., DINOv2 (Oquab et al., 2024)), and subsequent studies have focused on *when* and *how* to align. HASTE (Wang et al., 2025) identified a "capacity mismatch," suggesting that strict alignment becomes harmful in late training stages. iREPA (Singh et al., 2026) argued that the effectiveness of alignment stems from preserving spatial structure rather than global semantic information. While effective for images, these insights rely on the 2D spatial inductive bias of visual encoders, which does not directly translate to the temporal, sequential nature of audio spectral features.

Recent audio-domain adaptations have explored REPA-style alignment mainly in video-to-audio or Foley-generation settings. TARO (Ton et al., 2025) uses timestep-adaptive weights to align audio generation with video features, while HunyuanVideo-Foley (Shan et al., 2025) uses ATST-Frame features for alignment. Critically, these methods rely on frame- or clip-level feature signals as dense conditioning or alignment anchors. In our setting of token-conditioned audio synthesis, however, the model lacks such dense guidance and must infer continuous acoustics from sparse, discrete tokens. Moreover, existing REPA-style approaches typically rely on fixed or manually designed alignment locations or schedules, leaving the question of which internal audio-

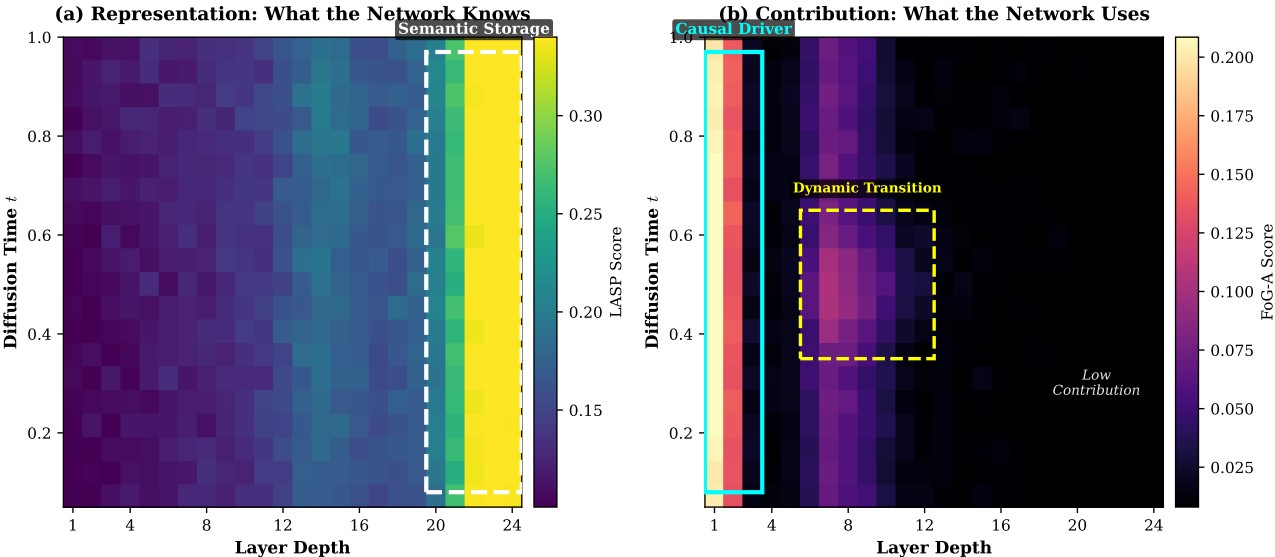

*Figure 1.* **The Spatiotemporal Anatomy of SCD.** We visualize the layer-wise dynamics across diffusion time steps ($t = 0 \rightarrow 1$). **(a) Representation:** LASP reveals teacher-dependent storage reservoirs: semantic storage is strongest in deep layers, while acoustic/event storage can shift toward middle layers under dense acoustic conditioning. **(b) Contribution:** In contrast, FoG-A reveals a dynamic functional landscape. Early layers (L1–3) act as strong **Causal Drivers**, and a **Dynamic Transition** emerges in the middle layers (L6–12) during the intermediate denoising phase ($t \approx 0.5$). **Key Insight:** The mismatch between teacher-similarity peaks and velocity-field attribution explains why fixed-depth alignment heuristics can be suboptimal.

generator layers should be aligned largely underexplored. Our work differs by proposing an *adaptive, data-driven* criterion for layer selection based on functional attribution.

### 2.3. Dissociation of Representation and Function

The distinction between "what a network represents" and "how it functions" has been increasingly recognized in interpretability research. Klabunde et al. (2025) surveyed metrics for representational similarity, cautioning that high similarity does not imply functional equivalence. Braun et al. (2025) provided an analytical treatment of this dissociation in deep linear networks, showing that functional and representational similarity can be decoupled even under controlled settings. Hase et al. (2023) similarly found that layers storing factual knowledge are not necessarily the most effective targets for model editing. We extend these findings to the domain of generative audio Flow Matching. By identifying that the layers most similar to the teacher (Representation) are not the ones driving the velocity field (Function), we provide the theoretical justification for why heuristic alignment methods are suboptimal for token-conditioned audio synthesis.

## 3. Mechanistic Insights

This section provides a theoretical lens for interpreting the empirical phenomena highlighted in the Introduction: *Store-Contribute Dissociation*, where information-rich layers do not necessarily coincide with causally critical layers. We analyze Token-conditioned FM by integrating the Information Bottleneck (IB) principle (Tishby & Zaslavsky, 2015) with a Neural Ordinary Differential Equation (ODE) perspective (Chen et al., 2018).

### 3.1. Decoupling Storage from Computation

*Information Bottleneck (IB)* views deep representations as trading off input compression and target-relevant prediction. However, in residual networks, skip connections allow information to persist in an additive stream even when intermediate transformations contribute weakly to the effective update (He et al., 2016). Empirically, residual architectures often rely disproportionately on short paths during training, with many longer paths contributing little gradient (Veit et al., 2016). This architectural property provides a mechanistic basis for the SCD: "what the network knows" (high mutual information) persists in the residual stream, while "what the network uses" (high gradient contribution) may be sparse and localized. This concept is related to prior findings showing that neural activity patterns may not reveal causal contributions (Fakhar et al., 2024), and that representational similarity does not imply functional similarity (Braun et al., 2025).

To formalize how information accumulates across layers and where computational contributions emerge, we adopt a dynamical systems view. Token-conditioned FM can be

cast as learning a time-dependent vector field for a continuous flow, whose sampling dynamics follow an ODE $d\mathbf{z}/dt = v_\theta(\mathbf{z}_t, t \mid X)$ (Lipman et al., 2023). For a fixed diffusion time $t$, the network that parameterizes $v_\theta(\cdot, t \mid X)$ is itself a depth-wise discrete dynamical system with residual updates, $\mathbf{z}_l = \mathbf{z}_{l-1} + f_l(\mathbf{z}_{l-1}, t, X)$ (He et al., 2016; Haber & Ruthotto, 2017). By telescoping,

$$\mathbf{z}_L = \mathbf{z}_0 + \sum_{k=1}^{L} f_k(\mathbf{z}_{k-1}, t, X), \qquad (1)$$

where each update depends on the evolving state $\mathbf{z}_{k-1}$. This formulation makes explicit the distinction between what each layer *accumulates* in the residual stream ($\mathbf{z}_l$) versus what each layer *contributes* ($f_l$), providing the formal basis for the SCD we observe empirically.

### 3.2. Sensitivity of Early Layers

The residual structure shown above suggests that perturbations introduced at earlier layers can influence a larger portion of the subsequent computation than perturbations introduced near the output. Let $\mathbf{J}_{k \to L} = \partial \mathbf{z}_L / \partial \mathbf{z}_k$ denote the Jacobian of the output with respect to the depth-$k$ state. Under the residual recursion, a perturbation at $k = 1$ propagates through the entire composition:

$$\delta \mathbf{z}_L = \mathbf{J}_{1 \to L} \, \delta \mathbf{z}_1, \qquad \mathbf{J}_{1 \to L} = \prod_{i=2}^{L} \left( \mathbf{I} + \frac{\partial f_i}{\partial \mathbf{z}_{i-1}} \right). \quad (2)$$

This yields a depth-wise "Butterfly Effect": perturbations introduced at Layer 1 are transformed by the full downstream Jacobian product $\mathbf{J}_{1 \to L}$ and can therefore affect all subsequent residual updates. However, the magnitude and direction of this effect depend on the spectra and orientations of the intervening Jacobian factors; the product may amplify, preserve, or attenuate a given perturbation. Thus, this analysis should be viewed as a mechanistic motivation rather than a proof that early layers are always more sensitive. It provides grounding for the SCD: although deep layers may *accumulate* richer representations, earlier layers can have high functional leverage because their updates are propagated through more downstream transformations. This motivates our use of FoG-A (Sec. 4.3) to empirically identify high-leverage layers, and informs the design of Attribution-Guided REPA (Sec. 4.4), which targets functionally critical layers rather than heuristically chosen ones.

### 3.3. Empirical Verification: Spatiotemporal Dynamics

To validate our theoretical model, we visualize the complete spatiotemporal landscape of the SCD in Figure 1. This heatmap contrasts the layer-wise Representation score (LASP) against the Contribution score (FoG-A) across the entire diffusion process ($t \in [0, 1]$).

As observed in Figure 1(a), the representational structure exhibits a time-invariant pattern. The deep layers (L20–24) consistently maintain high LASP scores (yellow region) regardless of the diffusion time step. This confirms that these layers act as a stable "Semantic Storage," holding the target acoustic information throughout the generation process.

The heatmap reveals a three-regime structure that is consistent with the mechanistic motivation in Section 3.2. The earliest layers (L1–3) form a continuous vertical "hot streak" across all timesteps, indicating consistently high FoG-A attribution to the velocity field. Crucially, the heatmap exposes a phenomenon invisible to static representational metrics: a *Dynamic Transition* region (dashed yellow box) emerges in the middle layers (L6–12), but only during intermediate timesteps ($t \approx 0.4 \to 0.7$), suggesting a functional handoff where mid-level refinements become important during specific phases of denoising. In contrast, Figure 1(b) shows that the deep layers (L16–24), despite being the most semantically rich in Panel (a), exhibit low FoG-A attribution, providing further evidence for the SCD.

This visual evidence suggests a limitation of fixed-depth REPA strategies. A static alignment at a manually chosen layer may miss layers with higher functional attribution at different denoising stages, while aligning deep layers may target information that the model stores but does not strongly use for velocity estimation. This motivates the attribution-guided approach we propose in Section 4.4.

## 4. Methodology

To operationalize the theoretical framework introduced in Section 3, we develop a unified interpretability toolkit comprising three complementary diagnostics: Bi-Stream Teacher Cosine Alignment (BiT-C), Layer-wise Analysis via Shared Projection (LASP), and Forward-only Gate Ablation (FoG-A).

Complementing these representational probes, FoG-A introduces an interventional metric to quantify functional necessity independent of information storage. As illustrated in Figure 2, we first employ BiT-C and LASP to diagnose *what networks know* (Representation Storage). Complementing this, Figure 3 demonstrates how FoG-A identifies *what networks use* (Causal Contribution), allowing AG-REPA to target functionally critical layers. This combined view directly exposes the SCD and forms the mechanistic basis of AG-REPA.

We focus on the practically relevant setting of **unified audio generation**, where a single model is trained to synthesize both speech and general audio. The overall framework is illustrated in Figure 4, with detailed architectural specifications provided in Appendix A.

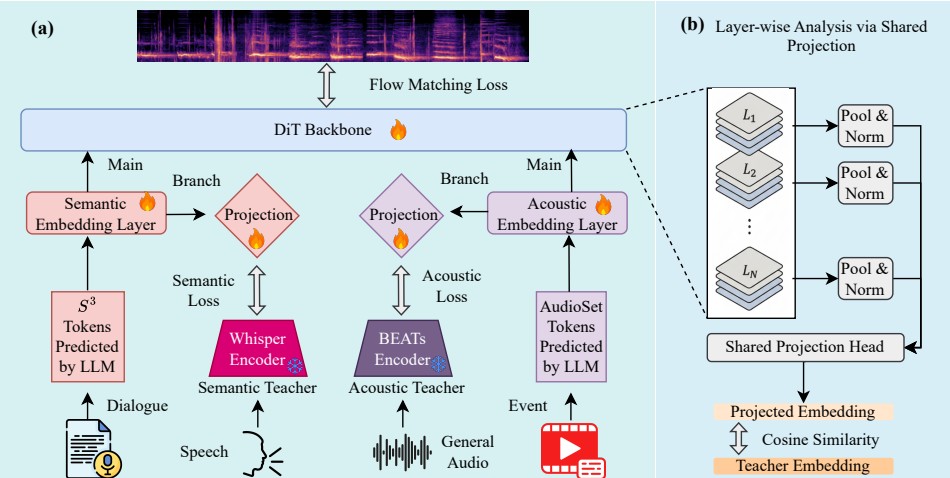

*Figure 2.* **Diagnosing Representation Storage. (a) BiT-C** establishes a dual-modality supervision baseline using frozen Whisper (semantic) and BEATs (acoustic) teachers to anchor the conditioning interface. **(b) LASP** probes "what the network knows" by projecting layer-wise representations into a shared teacher space using a frozen head, enabling cross-layer comparison of information storage.

## 4.1. Bi-Stream Teacher Cosine Alignment

Token-conditioned Flow Matching models process heterogeneous conditioning signals, including semantic tokens from speech encoders and acoustic tokens from general audio encoders, yet existing probing methods typically employ a single teacher reference. To comprehensively characterize layer-wise representations, we require alignment signals from both modalities.

BiT-C establishes a dual-teacher distillation framework that simultaneously aligns representations with two frozen teacher encoders: (i) the **Whisper Teacher** ($\mathcal{T}_{\text{speech}}$), a frozen Whisper large-v3 (Radford et al., 2023) encoder providing semantic supervision for speech-domain samples; and (ii) the **BEATs Teacher** ($\mathcal{T}_{\text{audio}}$), a frozen BEATs (Chen et al., 2023) encoder providing acoustic-level supervision for general audio samples. Given an intermediate representation $h_l \in \mathbb{R}^{B \times T \times D}$ at layer $l$, we first apply temporal mean pooling followed by layer normalization to obtain a global descriptor $\bar{h}_l \in \mathbb{R}^{B \times D}$:

$$\bar{h}_l = \text{LayerNorm}\left(\frac{1}{T}\sum_{t=1}^{T} h_l^{(t)}\right). \qquad (3)$$

A teacher-specific projection head $P_{\text{speech}} : \mathbb{R}^D \to \mathbb{R}^{D_{\text{teacher}}}$ then maps this descriptor into the teacher's embedding space, where alignment is measured via cosine similarity:

$$\text{BiT-C}_l^{\text{speech}} = \frac{\langle P_{\text{speech}}(\bar{h}_l), \mathcal{T}_{\text{speech}}(x)\rangle}{\|P_{\text{speech}}(\bar{h}_l)\| \cdot \|\mathcal{T}_{\text{speech}}(x)\|}, \qquad (4)$$

and analogously for $\text{BiT-C}_l^{\text{audio}}$.

During training, we optionally apply BiT-C as an auxiliary distillation objective only at the token-embedding output (input interface) to anchor the conditioning space without directly constraining deeper dynamics:

$$\mathcal{L}_{\text{BiT}} = \mathbb{E}_x\left[1 - \text{BiT-C}_0^{\text{speech}}\right] + \mathbb{E}_x\left[1 - \text{BiT-C}_0^{\text{audio}}\right]. \qquad (5)$$

During probing (evaluation), BiT-C runs in `no_grad` mode with frozen teachers and projection heads, reporting $\text{BiT-C}_l$ across depths $l$ to form the semantic/acoustic alignment curves.

## 4.2. Layer-wise Analysis via Shared Projection

LASP addresses a fundamental question: *What does each layer "know" about the target acoustic structure?* Unlike gradient-based attribution methods that measure sensitivity, LASP directly quantifies the informational content of layer representations.

A naive approach would train independent projection heads for each layer. However, this precludes meaningful cross-layer comparison, different heads may converge to different local optima, making their similarity scores incommensurate. LASP resolves this by employing a shared and frozen projection head (per teacher) across all layers, so that all depths are evaluated in a common metric space. Concretely, we reuse the teacher-specific projection heads $P_{\text{speech}}$ and $P_{\text{audio}}$ (trained at the input interface when BiT-C is enabled) and freeze them for probing. For each layer $l \in \{1, \dots, L\}$, we compute:

$$\text{LASP}_l^{(d)} = \mathbb{E}_{x \sim \mathcal{D}}\left[\cos\left(P_d(\bar{h}_l^{(d)}), \mathcal{T}_d(x)\right)\right], \qquad (6)$$

where $d \in \{\text{speech}, \text{audio}\}$. where $\bar{h}_l^{(d)}$ uses the same pooling as BiT-C.

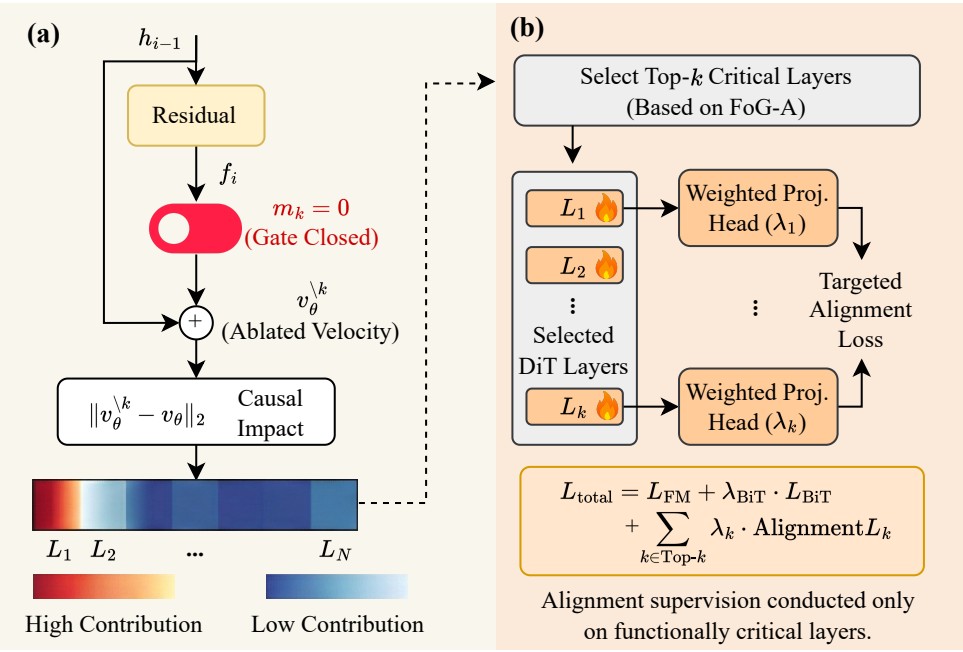

*Figure 3.* **From Causal Attribution to Optimization. (a) FoG-A** determines "what the network uses" by measuring the velocity field perturbation ($\|v_\theta^{\backslash k} - v_\theta\|$) caused by ablating individual layers, creating a functional attribution map. **(b) AG-REPA** utilizes the causal insights from FoG-A to selectively apply alignment supervision only to critical layers, bridging the gap between representation storage and causal contribution identified as the SCD.

## 4.3. Forward-only Gate Ablation

While LASP measures *what* each layer knows, it does not reveal *how much* each layer contributes to the final output. FoG-A addresses this orthogonal question: *What is the causal effect of removing a layer on the predicted velocity field?*

Let the DiT forward pass be defined by a sequence of residual updates:

$$h_l = h_{l-1} + m_l \cdot f_l(h_{l-1}, t, c), \qquad (7)$$

where $m_l \in \{0, 1\}$ acts as a computational gate. The standard model corresponds to $m_l = 1, \forall l$. We define the ablated velocity field $v_\theta^{\backslash k}$ by setting the gate $m_k = 0$ (intervention) while keeping $m_{j \neq k} = 1$:

$$v_\theta^{\backslash k}(x_t, t, c) \triangleq v_\theta(x_t, t, c)\Big|_{m_k=0}. \qquad (8)$$

The FoG-A score measures the normalized deviation caused by this intervention:

$$\text{FoG-A}_k = \mathbb{E}_{x_t, t, c}\left[ \frac{\left\| v_\theta^{\backslash k} - v_\theta \right\|_2}{\|v_\theta\|_2 + \epsilon} \right], \qquad (9)$$

where $\epsilon$ is a stability term. This metric directly quantifies the causal necessity of layer $k$.

## 4.4. Attribution-Guided REPA

The preceding analyses establish a fundamental asymmetry: while LASP and BiT-C identify which layers store semantic or acoustic information, FoG-A reveals which layers causally contribute to the output, and these two sets need not coincide (Store-Contribute Dissociation).

This insight motivates a principled refinement of standard REPA (Yu et al., 2025). REPA aligns at a single empirically-chosen layer (commonly layer 8; layer 4 for smaller models) with a global loss weight $\lambda$. While the authors hypothesize this allows deeper layers to focus on high-frequency details, this selection relies primarily on empirical tuning and does not account for distributed critical layers. Functionally critical layers can appear at varying depths depending on tokenizer design, and multiple layers may exhibit high causal importance simultaneously. To address this limitation, we propose Attribution-Guided REPA (AG-REPA), which (i) automatically selects Top-K layers based on causal attribution scores rather than a fixed single-layer choice, and (ii) assigns attribution-weighted regularization strength to each selected layer, enabling fine-grained control over the alignment process.

We utilize the pre-computed attribution scores $\mathcal{A} = \{\text{FoG-A}_k\}_{k=1}^L$ to construct a sparse layer set $\mathcal{S} \subset \{1, \ldots, L\}$ containing the top-$K$ layers ranked by attribution:

$$\mathcal{S} = \text{argtop}_K(\mathcal{A}). \qquad (10)$$

For each selected layer $k \in \mathcal{S}$, we assign an alignment weight $\lambda_k$ proportional to its FoG-A score:

$$\lambda_k = \frac{\text{FoG-A}_k}{\sum_{j \in \mathcal{S}} \text{FoG-A}_j}. \qquad (11)$$

This ensures that layers with higher causal contribution receive stronger alignment signals.

Each selected layer $k \in \mathcal{S}$ is equipped with a dedicated projection head $h_{\phi_k} : \mathbb{R}^D \to \mathbb{R}^{D_{\text{teacher}}}$, implemented as a two-layer MLP applied to the temporally-pooled descriptor $\bar{h}_k$; localized 1D/2D convolutional alternatives are compared in Appendix B and underperform the pooled MLP since the alignment target is itself a globally-pooled teacher embedding. The final training objective combines the Flow Matching loss and the BiT-C alignment loss with the sparse, weighted alignment penalty:

$$\mathcal{L}_{\text{total}} = \mathcal{L}_{\text{FM}} + \lambda_{\text{BiT}} \cdot \mathcal{L}_{\text{BiT}}$$
$$+ \sum_{k \in \mathcal{S}} \lambda_k \cdot \left(1 - \cos(h_{\phi_k}(\bar{h}_k), \mathcal{T}(x))\right), \qquad (12)$$

where $\mathcal{L}_{\text{BiT}}$ denotes the input interface alignment loss, $\bar{h}_k$ is the temporally-pooled representation at layer $k$, and $\mathcal{T}(x)$ is the frozen teacher embedding. By grounding both the layer selection $\mathcal{S}$ and the weighting $\lambda_k$ in causal attribution, AG-REPA explicitly targets the "Butterfly Effect" layers identified in Section 3.2. The set $\mathcal{S}$ is determined once via a probe-then-intervene protocol (Section A.5); we show in Appendix B that the FoG-A ranking is highly stable across epochs, so this static selection is a favorable efficiency–accuracy trade-off rather than a brittle assumption.

# 5. Results

Our Flow Matching models are trained on a combination of LibriSpeech (Panayotov et al., 2015) (speech domain) and AudioSet (Gemmeke et al., 2017) (general audio domain), following standard practice for unified audio generation evaluation. We compare two token topologies introduced in Sec. A.2: Config A ($S^3$ token (Du et al., 2024) + AudioSet token) and Config B (Config A further interleaving a dense BEATs prior (Chen et al., 2023) in a 1:1 manner). We quantify representation storage using LASP cosine probes (Cos-SEM / Cos-EVT; Sec. 4.2) and measure functional computation via FoG-A (Sec. 4.3). Table 1 summarizes the layer localization patterns by reporting the top-3 layers under each metric. Objective metrics (WER, FAD) are reported as three-seed means at the fixed 500k-step checkpoint, while MOS is reported as mean±standard error over human ratings.

## 5.1. Quantitative Verification of Dissociation

Our IB-ODE framework (Sec. 3) predicts that information storage and causal contribution need not be co-localized.

The results support this non-co-localization. Cos-SEM peaks are concentrated in deep layers, while Cos-EVT is topology dependent: it is late-layer dominated in Config A but shifts to middle layers under BEATs interleaving in Config B. In contrast, FoG-A reveals that functional necessity is dominated by early bottlenecks, especially Layer 1 for both speech (FoG-A$_1$ = 0.167) and audio (FoG-A$_1$ = 0.140), with additional middle-layer contributors under Config B. Thus, SCD should be understood as a separation between teacher-similarity peaks and velocity-field attribution peaks, rather than a universal claim that all deep layers are inactive. This decoupling between "what the network knows" and "what the network uses" forms the mechanistic basis of AG-REPA.

## 5.2. Attribution-Guided vs. Static Alignment

Table 2 compares **AG-REPA** against static REPA baselines (Yu et al., 2025) applied to fixed layers. Static variants (Layers 4, 8, 12) yield only marginal gains, precisely because these layers lie outside the causal-dominant regime identified by FoG-A. AG-REPA instead supervises the functionally critical bottlenecks: the input interface ($L_1$) where Jacobian sensitivity is maximal and the other top-2 functional attribution layers. This yields FAD reductions of **18%** (speech) and **16%** (audio) over the best single-layer baseline, and further outperforms the multi-layer heuristic (REPA @ Layers 4,8,12) by **11%**, confirming that aligning *what the network uses*, rather than what it merely stores, is essential for maximizing representation alignment efficacy. Moreover, this precision translates to superior perceptual quality, with AG-REPA achieving the lowest WER (3.45) and highest MOS (4.12), indicating that focusing on causal bottlenecks effectively enhances both intelligibility and naturalness without compromising generative diversity. The MOS gain over the strongest fixed mid-layer baseline (REPA @ L4,8,12) is statistically significant: $p = 0.004$ (speech) and $p = 0.011$ (audio) under a paired two-sided $t$-test, with 95% CIs $[4.02, 4.22]$ (speech) and $[3.80, 4.08]$ (audio) for AG-REPA.

Beyond the fixed mid-layer baselines, Table 2 also includes two depth-diagnostic controls. *Deep REPA* aligns a representation-rich band (L20–L22) near the LASP peaks (Sec. 4.2), whereas *Shallow REPA* aligns the early blocks (L1–L3) that dominate FoG-A attribution (Sec. 4.3). Consistent with the SCD (Sec. 5.1), Deep REPA yields only marginal gains over the no-intermediate-layer-alignment baseline despite strong teacher similarity. In contrast, Shallow REPA recovers most of the improvement by directly supervising the causal bottleneck near the input interface. Finally, AG-REPA improves upon Shallow REPA by *sparsely selecting and adaptively weighting* the most highly attributed layers, avoiding over-constraining the early stack while better preserving generative flexibility.

*Table 1.* **Store-Contribute Dissociation.** We report the top-3 layers for representation *storage* (Cos-SEM / Cos-EVT from LASP probes, denoting alignment with Whisper semantic and BEATs acoustic event teachers, respectively) and functional *contribution* (FoG-A sensitivity) on LibriSpeech (Speech) and AudioSet (Audio).

| Config | Cos-SEM (top3) | Cos-EVT (top3) | FoG-A Speech (top3) | FoG-A Audio (top3) |
|---|---|---|---|---|
| **A: $S^3$ + AudioSet** | $L_{24}(0.323)$ $L_{18}(0.293)$ $L_{17}(0.293)$ | $L_{20}(0.180)$ $L_{21}(0.174)$ $L_{19}(0.168)$ | $L_1(0.167)$ $L_9(0.0695)$ $L_5(0.0588)$ | $L_1(0.140)$ $L_{21}(0.060)$ $L_9(0.0561)$ |
| **B: A + interleaved BEATs** | $L_{23}(0.318)$ $L_{22}(0.314)$ $L_{24}(0.309)$ | $L_{14}(0.218)$ $L_{13}(0.212)$ $L_{15}(0.207)$ | $L_1(0.192)$ $L_2(0.109)$ $L_7(0.085)$ | $L_1(0.164)$ $L_7(0.094)$ $L_2(0.083)$ |

*Table 2.* **Comparison of Alignment Strategies.** Evaluation of intelligibility (WER ↓), generation quality (FAD ↓) and perceptual fidelity (MOS ↑) under Config B. The baseline retains only the input-interface BiT-C anchor and uses no intermediate-layer REPA; **AG-REPA** achieves the best performance by sparsely selecting and adaptively weighting functionally critical layers.

| Method | Speech WER ↓ | Speech FAD ↓ | Audio FAD ↓ | Speech MOS ↑ | Audio MOS ↑ |
|---|---|---|---|---|---|
| Base (no layer align.) | 5.82 | 1.84 | 3.45 | 3.62±.08 | 3.45±.09 |
| REPA @ Layer 4 | 5.15 | 1.65 | 3.12 | 3.75±.07 | 3.58±.08 |
| REPA @ Layer 8 | 4.93 | 1.58 | 3.05 | 3.79±.06 | 3.64±.08 |
| REPA @ Layer 12 | 5.38 | 1.72 | 3.28 | 3.68±.08 | 3.51±.09 |
| REPA @ L4, 8, 12 | 4.21 | 1.45 | 2.88 | 3.92±.06 | 3.77±.07 |
| REPA @ Deep (L20–L22) | 5.60 | 1.79 | 3.39 | 3.64±.08 | 3.48±.09 |
| REPA @ Shallow (L1–L3) | 3.62 | 1.36 | 2.68 | 4.05±.06 | 3.87±.07 |
| **AG-REPA (Top-3)** | **3.45** | **1.29** | **2.56** | **4.12±.05** | **3.94±.07** |

## 5.3. The Impact of Alignment Targets

This ablation examines whether alignment is more effective when applied to layers that are most *similar* to the teacher ("knowing", high LASP) or to layers that are most *necessary* for the velocity field ("doing", high FoG-A). Table 3 contrasts four selection rules under the same training budget.

Two controls are informative. First, random layers bring only a minor improvement over the baseline and do not speed up optimization, suggesting that the gains are not a generic regularization effect. Second, choosing the top LASP layers, which sit in the representation-rich deep regime, helps more than random, but the improvement remains modest.

In contrast, Gradient Norm selects layers with the highest loss-gradient magnitude under $\mathcal{L}_{FM}$. It is therefore a practical proxy for contribution, reflecting where the optimizer concentrates its updates. FoG-A is more mechanistic: it evaluates contribution via forward ablation and measures the induced change in the velocity field $v_\theta$. Consistent with this distinction, Table 3 shows that Gradient Norm improves over LASP-based, storage-driven selection, but remains slightly behind FoG-A in both final Speech/Audio FAD (1.35/2.71 vs. 1.29/2.56) and convergence speed. Grad-

Norm selects $L_4$ (high gradient magnitude but weak causal effect on $v_\theta$) and misses the mid-phase transition bottleneck $L_7$ revealed by FoG-A in Figure 1(b); the FoG-A probe itself takes $< 0.5\%$ of total training wall-clock (Section A.5).

In summary, these results support the paper's central claim: alignment is most effective when applied to the layers that the model uses to form $v_\theta$, rather than to those that merely store teacher-aligned features.

## 5.4. Generalization of AG-REPA

A practical concern is whether attribution-guided alignment is specific to our unified DiT-based setup, or whether it transfers to other Flow Matching systems with different architectures and training recipes. To probe this, we apply AG-REPA to three representative models (Voicebox (Le et al., 2023), CosyVoice (Du et al., 2024), and F5-TTS (Chen et al., 2025)) while keeping each baseline unchanged. For each model, we run FoG-A to obtain a layer-wise attribution profile, then apply alignment only to the causally dominant layers identified by that profile.

Table 4 shows that AG-REPA yields consistent improvements across all three architectures. On Voicebox, we observe a clear quality shift, with FAD dropping from 1.20 to 0.95 and WER improving from 2.05 to 1.85. On the strong CosyVoice baseline, the gains are smaller but still steady across all metrics (WER $1.95 \rightarrow 1.78$, FAD $0.88 \rightarrow 0.72$, MOS $4.25 \rightarrow 4.39$), indicating that attribution-guided supervision does not trade off intelligibility for perceptual quality. Finally, for F5-TTS, AG-REPA improves both objective and subjective metrics (FAD $1.45 \rightarrow 1.15$, MOS $4.05 \rightarrow 4.22$).

A direct concern is whether the Shallow REPA heuristic (hard-coded $L_1$–$L_3$), which is already competitive on our DiT (Table 2), might capture most of AG-REPA's gain on other architectures. Table 5 answers this directly. While Shallow REPA is strong on the architecture for which the heuristic was calibrated, its advantage erodes on other backbones because the FoG-A peak location varies with tokenizer design and depth. On F5-TTS, for instance, Shallow REPA recovers only a 7.6% FAD gain ($1.45 \rightarrow 1.34$), whereas AG-REPA achieves a 20.7% gain ($1.45 \rightarrow 1.15$)

*Table 3.* **The Impact of Alignment Targets: Knowing vs. Doing.** Aligning functionally critical layers (identified by FoG-A) yields $3.4\times$ larger FAD reduction than aligning representation-rich layers (identified by LASP), and accelerates convergence by $3.3\times$, directly validating the SCD. Audio FAD is shown for all selectors; the measured Gradient Norm vs. FoG-A comparison highlights that FoG-A also dominates on the general-audio domain.

| Selection Strategy | Top-3 Layers | Training Steps | Speech FAD ↓ | Audio FAD ↓ | Rel. Gain | Convergence[†] |
|---|---|---|---|---|---|---|
| Base (no layer align.) | NONE | 500k | 1.84 | 3.45 | 0.0% | N/A |
| Random Control | $L_5, L_{14}, L_{19}$ | 500k | 1.75 | 3.32 | +4.9% | 850k |
| Highest LASP | $L_{22}, L_{23}, L_{24}$ | 500k | 1.68 | 3.21 | +8.7% | 720k |
| Gradient Norm | $L_1, L_2, L_4$ | 500k | 1.35 | 2.71 | +26.6% | 260k |
| **Highest FoG-A (Ours)** | $L_1, L_2, L_7$ | 500k | **1.29** | **2.56** | **+29.9%** | **220k** |

[†]Convergence reports the number of optimization steps on the same extended training curve required to reach Speech FAD = 1.5; it is therefore independent of the fixed 500k checkpoint used for the FAD columns. N/A indicates that the no-intermediate-layer-alignment baseline is used only as the reference and did not reach the threshold within the evaluated continuation window.

*Table 4.* **Generalization of AG-REPA.** Comparison of standard training (Baseline) versus our method across different Flow Matching architectures. AG-REPA consistently improves intelligibility (WER), audio quality (FAD), and naturalness (MOS).

| Model | WER ↓ | FAD ↓ | MOS ↑ |
|---|---|---|---|
| Voicebox (Le et al., 2023) | 2.05 | 1.20 | 4.15 ± .06 |
| **+ AG-REPA** | **1.85** | **0.95** | **4.28 ± .05** |
| CosyVoice (Du et al., 2024) | 1.95 | 0.88 | 4.25 ± .05 |
| **+ AG-REPA** | **1.78** | **0.72** | **4.39 ± .04** |
| F5-TTS (Chen et al., 2025) | 2.40 | 1.45 | 4.05 ± .07 |
| **+ AG-REPA** | **2.12** | **1.15** | **4.22 ± .06** |

*Table 5.* **Shallow REPA vs. AG-REPA Across Architectures (FAD ↓).** AG-REPA transfers better by re-running FoG-A per backbone.

| Architecture | Base. | Shallow REPA | AG-REPA |
|---|---|---|---|
| Our DiT | 1.84 | 1.36 | **1.29** |
| F5-TTS (Chen et al., 2025) | 1.45 | 1.34 | **1.15** |
| Voicebox (Le et al., 2023) | 1.20 | 1.12 | **0.95** |
| CosyVoice (Du et al., 2024) | 0.88 | 0.85 | **0.72** |

The causal layer set $\mathcal{S}$ is highly stable across training: re-probing every epoch yields a Top-3 overlap of $2/3$–$3/3$ with the warm-up selection, and continuously refreshing $\mathcal{S}$ changes Speech/Audio FAD only marginally ($1.29/2.56 \to 1.28/2.55$) while inflating wall-clock by $+19\%$. Ablations in Appendix B further establish that $K = 3$ with attribution-proportional weights is the optimal trade-off, that timestep-adaptive routing brings no gain over the static selection, and that pooled 2-layer MLP projection heads outperform 1D/2D convolutional alternatives.

by re-running the diagnostic per architecture; similar margins hold for Voicebox and CosyVoice. This confirms that the value of FoG-A is precisely its ability to adapt to each model's intrinsic causal topology rather than commit to a fixed depth band.

Overall, these results indicate that the "knowing vs. doing" gap is not specific to a particular architecture. Across different Flow Matching systems, teacher similarity does not necessarily identify the layers that govern generation dynamics, and FoG-A offers a reliable criterion for locating the layers where alignment is most effective.

## 6. Conclusion

We uncover the **Store-Contribute Dissociation** in token-conditioned audio Flow Matching and accordingly propose **Attribution-Guided REPA**, which uses FoG-A causal attribution rather than depth-based heuristics to align functionally critical layers, improving generation quality while accelerating convergence. Our findings show that knowing is not doing: efficient generative training should be guided not only by where teacher similarity is high, but by where representations are causally consequential for the velocity field. The evidence in this work suggests that such storage-contribution separation may extend beyond our backbone to other residual generative models, although its exact layer locations depend on token topology and architecture. While the causal layer set remains stable in our setting, dynamic refresh may be worth revisiting for longer runs or architectures with stronger representation drift.

### 5.5. Robustness and Efficiency of Causal Selection

This subsection consolidates the core robustness findings; full ablations are deferred to Section A.5 and Appendix B.

Although FoG-A is an interventional diagnostic, its wall-clock cost is negligible because it executes only during a one-time warm-up epoch with sparse, gradient-free probes (every 200 steps, mini-batch $\leq 2$, single GPU, `torch.no_grad()`). The diagnostic adds $< 0.5\%$ to total wall-clock time, preserving an end-to-end convergence speedup of $\approx 3.26\times$ over LASP-selected REPA (Table 6). During main training, AG-REPA further adds only $K = 3$ lightweight MLP projection heads ($< 0.5\%$ extra parameters, $< 2\%$ per-step wall-clock vs. standard REPA).

## Impact Statement

This paper introduces **AG-REPA**, a framework designed to enhance both the generation quality and the transparency of Flow Matching models for audio synthesis. Our work contributes primarily to the methodological advancement of generative machine learning by proposing a unified interpretability toolkit to "unlock the black box" of audio generation models.

**Potential Benefits.** By significantly reducing Word Error Rate (WER) and improving perceptual quality through targeted alignment of causal bottlenecks, our work directly enables more intelligible and robust unified audio synthesis systems. Crucially, our contribution extends beyond performance metrics; we establish a **comprehensive interpretability Toolkit** (comprising BiT-C, LASP, and FoG-A) that decouples representation storage from causal contribution. We specifically highlight the interpretability value of AG-REPA: it serves as a proof-of-concept that mechanistic insights can be directly operationalized into superior training strategies. By shifting from heuristic choices to *attribution-guided optimization*, our work paves the way for building generative AI systems that are not only more efficient but also scientifically grounded, transparent, and controllable.

**Limitations.** Our probe-then-intervene protocol is intentionally static after the warm-up segment; this is efficient and empirically stable in our 500k-step setting, but longer training runs or architectures with stronger representation drift may benefit from occasional re-probing. In addition, our pooled MLP projection head is well matched to globally-pooled Whisper/BEATs targets, but localized convolutional heads may become preferable when the teacher signal preserves dense temporal or spatial structure.

**Risks and Mitigations.** We acknowledge that advancements in high-fidelity audio generation, particularly those involving few-shot style transfer or voice cloning, carry inherent risks of misuse. These include the potential for creating misleading content (deepfakes), impersonation, and voice spoofing. While our work focuses on the training dynamics rather than releasing a new large-scale foundation model, we advocate for the responsible deployment of such technologies. Future applications building on this work should incorporate safeguards such as audio watermarking, spoofing detection mechanisms, and restricted access to voice cloning capabilities to mitigate these societal risks.

## Acknowledgements

This work was supported by the National Natural Science Foundation of China (No. 62471420), Guang-dong Basic and Applied Basic Research Foundation (2025A1515012296), and 2025 Tencent AI Lab Rhino-Bird Program.

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

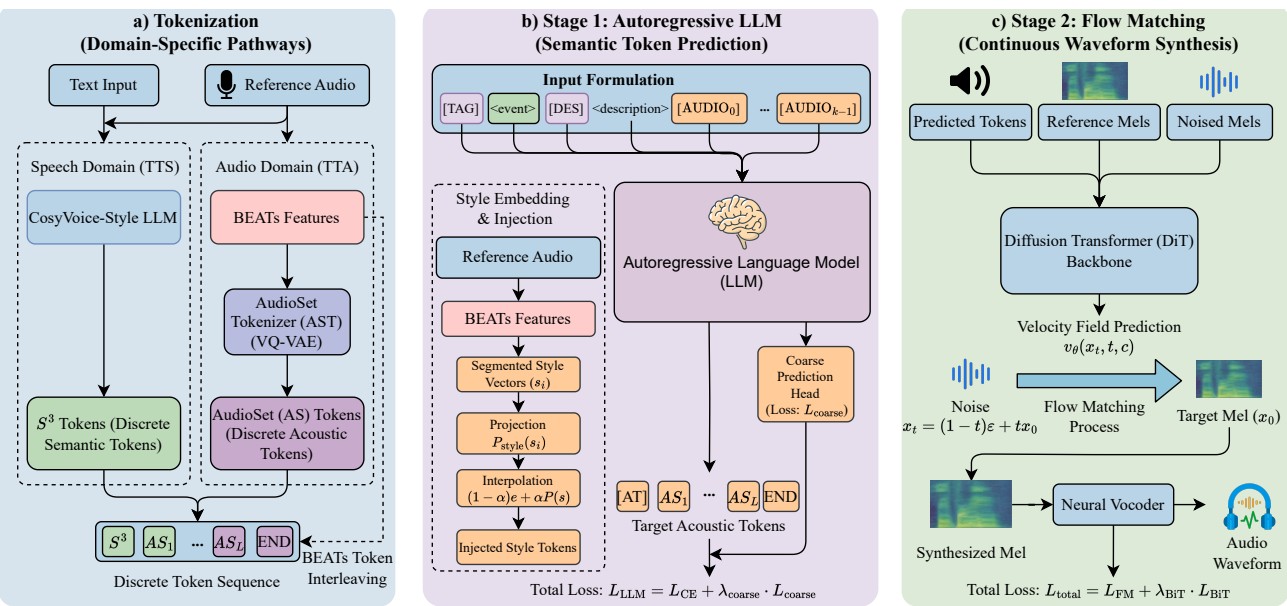

*Figure 4.* **The Unified Audio Generation Framework.** The system utilizes a two-stage cascade architecture decoupling semantic planning from acoustic rendering: **(a) Tokenization:** Domain-specific pathways process inputs into a unified discrete sequence, utilizing $S^3$ tokens for speech and AudioSet (AS) tokens for audio, optionally interleaved with BEATs features to inject dense acoustic priors. **(b) Stage 1 (Autoregressive LLM):** A causal language model predicts target acoustic tokens, incorporating reference style via a learnable projection and injection module ($P_{\text{style}}$) while utilizing an auxiliary coarse prediction head. **(c) Stage 2 (Flow Matching):** A Diffusion Transformer (DiT) backbone predicts the velocity field $v_\theta$ to transform noise into target mel-spectrograms via Flow Matching, conditioned on projected tokens and reference audio, before final waveform synthesis by a neural vocoder.

## A. Framework

Building upon the theoretical insights presented in Section 3, we now describe our unified audio generation framework, which consists of a two-stage cascade architecture: an autoregressive Large Language Model (LLM) for acoustic token prediction, followed by a Flow Matching model for continuous waveform synthesis. This design decouples high-level semantic planning from low-level acoustic rendering, enabling flexible conditioning across both speech and general audio domains.

### A.1. System Overview

Figure 4 illustrates the complete pipeline. Given a text prompt (with optional event tags and descriptions for audio generation) and a reference audio clip, the system operates in two stages:

1. **Stage 1 (LLM)**: An autoregressive language model predicts a sequence of discrete acoustic tokens conditioned on text and style embeddings extracted from the reference audio.
2. **Stage 2 (Flow Matching)**: A Diffusion Transformer (DiT) based Flow Matching model transforms the predicted tokens into continuous mel-spectrograms, which are subsequently decoded to waveforms via the Vocos neural vocoder (Siuzdak, 2024).

This cascade architecture enables unified handling of both Text-to-Speech (TTS) and Text-to-Audio (TTA) tasks through domain-specific tokenization pathways.

### A.2. Tokenization

Our framework employs domain-specific tokenizers to bridge the gap between discrete symbolic representations and continuous acoustic features:

**Speech Domain ($S^3$ Tokens).** For speech synthesis, we adopt the $S^3$ token vocabulary from CosyVoice (Du et al., 2024), which provides a semantically rich representation derived from self-supervised speech models. Given a text input and reference speech, we employ a CosyVoice-style LLM to predict the target $S^3$ token sequence in a zero-shot manner.

**Audio Domain (AudioSet Tokens).** For general audio generation, we trained a dedicated **AudioSet Tokenizer** (AST) based on the RepCodec VQ-VAE architecture (Huang et al., 2024). The AST processes BEATs (Chen et al., 2023) features and learns a discrete codebook of size $V = 4096$, aligned with the $S^3$ vocabulary for unified downstream processing:

$$\mathbf{z}_{\text{AS}} = \text{Quantize}\big(\text{Proj}(\text{Enc}(\mathbf{f}_{\text{BEATs}}))\big), \tag{13}$$

where $\mathbf{f}_{\text{BEATs}} \in \mathbb{R}^{T \times D}$ denotes the frame-level BEATs features. Training minimizes a combination of reconstruction loss and VQ commitment loss:

$$\mathcal{L}_{\text{AST}} = \mathcal{L}_{\text{recon}} + \beta \cdot \mathcal{L}_{\text{commit}}. \tag{14}$$

**BEATs Token Interleaving (Dual-Codebook Variant).** To inject dense acoustic priors into the Flow Matching model, we optionally employ a 1:1 interleaving strategy between primary tokens ($S^3$/AS) and BEATs tokens. Given a primary token sequence $\{t_1, t_2, \ldots, t_L\}$ and a corresponding BEATs token sequence $\{b_1, b_2, \ldots, b_L\}$, the interleaved sequence is:

$$\mathbf{s}_{\text{interleaved}} = [t_1, b_1, t_2, b_2, \ldots, t_L, b_L]. \tag{15}$$

This creates an intermediate proxy manifold $\mathcal{M}_{\text{prior}}$ that is topologically proximal to the target acoustic manifold, accelerating the FM training process while enhancing the generated audio quality.

### A.3. Stage 1: Autoregressive LLM with Style Injection

The first stage employs a pretrained causal language model, finetuned to predict acoustic tokens given textual and stylistic conditioning.

**Input Formulation.** For Text-to-Audio generation, the input sequence is constructed as:

$$\begin{aligned} &\texttt{[TAG]} \langle \text{event} \rangle \texttt{[DES]} \langle \text{description} \rangle \big\{ \texttt{[AUDIO}_i\texttt{]} \big\}_{i=0}^{K-1} \\ &\xrightarrow{\text{generate}} \texttt{[AT]} \big\{ \texttt{[AS}_j\texttt{]} \big\}_{j=1}^{L} \texttt{[END]} \end{aligned} \tag{16}$$

where $K$ style placeholder tokens $\texttt{[AUDIO}_i\texttt{]}$ are replaced with BEATs-derived style embeddings during training and inference.

**Style Embedding and Injection.** Given a reference audio, we extract $K$ temporally-segmented style vectors from BEATs features:

$$\mathbf{s}_i = \frac{1}{|T_i|} \sum_{t \in T_i} \mathbf{f}_{\text{BEATs}}^{(t)}, \quad i \in \{0, \ldots, K-1\}, \tag{17}$$

where $T_i$ denotes the $i$-th temporal segment. These are projected to the LLM hidden dimension via a learnable projection $P_{\text{style}} : \mathbb{R}^{D_{\text{BEATs}}} \to \mathbb{R}^H$, and injected via linear interpolation:

$$\tilde{\mathbf{e}}_{\texttt{[AUDIO}_i\texttt{]}} = (1 - \alpha) \cdot \mathbf{e}_{\texttt{[AUDIO}_i\texttt{]}} + \alpha \cdot P_{\text{style}}(\mathbf{s}_i), \tag{18}$$

where $\alpha$ controls the injection strength. During training, we apply style dropout with probability $p$ to enable classifier-free guidance at inference.

**Training Objective.** The LLM, initialized from Qwen3-0.6B-Base (Yang et al., 2025), is trained with a combined objective:

$$\mathcal{L}_{\text{LLM}} = \mathcal{L}_{\text{CE}} + \lambda_{\text{coarse}} \cdot \mathcal{L}_{\text{coarse}}, \tag{19}$$

where $\mathcal{L}_{\text{CE}}$ is the standard causal language modeling cross-entropy loss over the target token sequence, and $\mathcal{L}_{\text{coarse}}$ is an auxiliary coarse prediction loss. The coarse loss supervises a linear head to predict cluster assignments (derived from codebook structure), providing hierarchical guidance:

$$\mathcal{L}_{\text{coarse}} = \mathbb{E}_{t \in \mathcal{T}_{\text{AS}}} \big[ \text{CE}\big(h_\phi(\mathbf{h}_t), c(y_t)\big) \big], \tag{20}$$

where $h_\phi$ is the coarse prediction head, $\mathbf{h}_t$ is the hidden state at position $t$, and $c(y_t)$ maps the target token to its cluster ID.

## A.4. Stage 2: Flow Matching with DiT Backbone

The second stage employs a Diffusion Transformer (DiT) based Flow Matching model (Lipman et al., 2023) to transform discrete token embeddings into continuous mel-spectrograms.

**Conditioning Interface.** We define the conditioning signal $\mathbf{c}$ as the concatenation of the projected token embedding and the reference mel-spectrogram:

$$\mathbf{c} = [\mathbf{e}_{\text{fused}} \,\|\, \mathbf{m}_{\text{ref}}], \tag{21}$$

where $\mathbf{e}_{\text{fused}} \in \mathbb{R}^{D_{\text{tok}} \times T}$ is the projected token embedding (linearly interpolated to match mel resolution) and $\mathbf{m}_{\text{ref}} \in \mathbb{R}^{N_{\text{mel}} \times T}$ is the reference mel-spectrogram (zero-padded beyond the prompt region). At diffusion time $t$, the DiT input is formed by concatenating the noised mel $\mathbf{x}_t \in \mathbb{R}^{N_{\text{mel}} \times T}$ with the conditioning, i.e. $[\mathbf{x}_t \,\|\, \mathbf{c}]$; accordingly, the velocity field is written $v_\theta(\mathbf{x}_t, t, \mathbf{c})$ throughout.

**DiT Architecture.** The backbone consists of $L$ DiT blocks with adaLN-Zero conditioning (Peebles & Xie, 2023). Each block applies two sequential residual sub-updates, where the time embedding $\mathbf{t}$ drives both the adaptive layer norm and the zero-initialized residual gates $\alpha_l^{\text{attn}}, \alpha_l^{\text{ffn}}$:

$$
\begin{aligned}
\mathbf{h}_l' &= \mathbf{h}_{l-1} + \alpha_l^{\text{attn}}(\mathbf{t}) \cdot \text{Attn}_l\big(\text{AdaLN}(\mathbf{h}_{l-1}; \mathbf{t})\big), \\
\mathbf{h}_l &= \mathbf{h}_l' + \alpha_l^{\text{ffn}}(\mathbf{t}) \cdot \text{FFN}_l\big(\text{AdaLN}(\mathbf{h}_l'; \mathbf{t})\big),
\end{aligned}
\tag{22}
$$

where $\mathbf{t}$ is the sinusoidal time embedding and the gates $\alpha_l^{\text{attn}}, \alpha_l^{\text{ffn}}$ are initialized to zero, so that each block starts as the identity map (Peebles & Xie, 2023). Collapsing the two sub-updates into a single per-layer contribution $f_l(\cdot)$ recovers the residual form $\mathbf{h}_l = \mathbf{h}_{l-1} + f_l(\cdot)$ used in our mechanistic analysis (Section 3); the FoG-A gate $m_l$ (Equation (9)) ablates this entire contribution. The network predicts the velocity field $v_\theta(\mathbf{x}_t, t, \mathbf{c})$ for the optimal transport path from noise to data.

**Training Objective.** The primary objective is the Flow Matching loss:

$$\mathcal{L}_{\text{FM}} = \mathbb{E}_{t, \mathbf{x}_0, \boldsymbol{\epsilon}}\Big[\big\|v_\theta(\mathbf{x}_t, t, \mathbf{c}) - (\mathbf{x}_0 - \boldsymbol{\epsilon})\big\|_2^2\Big], \tag{23}$$

where $\mathbf{x}_t = (1 - t)\boldsymbol{\epsilon} + t\mathbf{x}_0$ and $\boldsymbol{\epsilon} \sim \mathcal{N}(0, \sigma^2 I)$.

This unified framework enables both TTS and TTA within a single model by switching tokenization pathways, while the Flow Matching stage remains shared across domains.

## A.5. Probe-then-Intervene Protocol, Efficiency, and Stability

To prevent potential concerns about "online" layer selection or teacher-induced bias, we adopt a strict *probe-then-intervene* schedule that cleanly separates **diagnosis** (layer probing) from **optimization** (alignment training). Concretely, all layer probes (Sections 4.1 to 4.3) are computed only once during an initial 5,000-step diagnostic warm-up segment, and the resulting layer choices are then *frozen*. The main FAD/WER/MOS tables are evaluated at a fixed 500k-step checkpoint, while convergence statistics are measured on the corresponding extended training curves.

**Phase I: Diagnostic warm-up (first 5,000 steps).** We first train the Flow Matching model with the standard objective (Flow Matching loss plus the *interface-level* BiT-C distillation, Section 4.1 and Equation (4)). During this warm-up segment, we run two complementary probes: (i) Representation probes (BiT-C / LASP) to identify which layers most strongly *store* semantic or acoustic teacher information; and (ii) Functional attribution (FoG-A, Section 4.3 and Equation (9)) to quantify which layers *causally affect* the predicted velocity field. All probes are executed with frozen teachers and in `no_grad` mode. At the end of the warm-up segment, we compute the Top-$K$ causal layer set $\mathcal{S}$ and attribution weights $\{\lambda_k\}_{k \in \mathcal{S}}$ based on the measured attribution gap, and *freeze* them for subsequent training.

**Phase II: Main training to the fixed 500k checkpoint.** Starting from the same post-warmup checkpoint, we branch into two runs that share identical data, optimization budget, and hyper-parameters:

1. **Baseline (Control).** We continue the original training procedure without any intermediate-layer alignment. The model is optimized with $\mathcal{L}_{\text{FM}}$ and retains the same *interface-level* BiT-C distillation $\mathcal{L}_{\text{BiT}}$ to ensure consistent conditioning anchoring.

2. **AG-REPA (Intervention).** We activate the sparse, attribution-weighted feature distillation term on the frozen causal layer set $\mathcal{S}$ (Section 4.4 and Equation (12)). Crucially, to ensure a clean causal intervention, we disable LASP probing hooks (no further layer-wise probing) and maintain the shared *interface-level* BiT-C loss. This isolates the performance gain solely to the attribution-guided intermediate alignment.

After the fixed 500k-step checkpoint, we compare the two branches on both speech and general audio synthesis. This protocol ensures that: (a) the causal selection signal is obtained once from the training set; (b) the selection is fixed during the 500k-step optimization window; and (c) the Baseline and AG-REPA differ *only* in the application of attribution-guided intermediate-layer supervision.

**Efficiency of FoG-A Diagnostics.** A natural concern is whether interventional layer probing imposes a non-trivial cost. We show that the FoG-A diagnostic is essentially free relative to the main training budget. Five design choices make this possible: **(i)** *Sparse temporal sampling*: FoG-A is triggered only every 200 optimization steps, not every step; **(ii)** *Small probing batch*: each call uses a mini-batch of $\leq 2$ samples (1 speech + 1 audio); **(iii)** *Layer alternation*: each call probes 12 of 24 layers via even/odd alternation, costing 13 forward passes (1 baseline + 12 ablated); **(iv)** *Rank-0 execution*: only one GPU runs the diagnostic; **(v)** *Gradient-free probing*: the entire FoG-A loop executes under `torch.no_grad()` with zero gradient memory. For the 5,000-step warm-up segment with training batch size $B = 16$, FoG-A is triggered 25 times, totaling $\approx 41$ equivalent training single-forward passes; since a normal training step includes both a forward and a more expensive backward pass, this adds $< 0.5\%$ to the total wall-clock time, yielding the end-to-end speedup of $\approx 3.26\times$ over LASP-selected REPA summarized in Table 6. The per-step training overhead is also small: AG-REPA adds only $K = 3$ lightweight two-layer MLP projection heads ($< 0.5\%$ extra parameters) and incurs $< 2\%$ per-step wall-clock overhead relative to standard REPA.

*Table 6.* **End-to-End Efficiency.** The FoG-A diagnostic adds $< 0.5\%$ wall-clock; the end-to-end speedup over LASP-selected REPA (the strongest storage-driven baseline reaching FAD = 1.5; Table 3) is $\approx 3.26\times$.

| Method | Main Training | Diagnostic | End-to-End |
|---|---|---|---|
| LASP-selected REPA | $T_0$ | $\approx 0$ | $1.00\,T_0$ |
| **AG-REPA (Ours)** | $\approx 0.30\,T_0$ | $< 0.005\,T_0$ | $\approx 0.305\,T_0$ |

**Cross-Checkpoint Stability of FoG-A Rankings.** The probe-then-intervene protocol freezes $\mathcal{S}$ after the warm-up segment. A natural concern is whether this static choice remains valid under representation drift. We therefore re-ran FoG-A at later training checkpoints and report the Top-3 rankings in Table 7. Layer 1 remains the dominant causal driver throughout training, consistent with the Jacobian sensitivity analysis in Section 3.2, and the Top-3 sets exhibit an overlap of 2/3–3/3 with the warm-up selection across both modalities. Continuously refreshing $\mathcal{S}$ yields only marginal FAD changes ($1.29/2.56 \rightarrow 1.28/2.55$) while adding $+19\%$ probe cost (Table 8). The static selection is therefore a favorable efficiency–accuracy trade-off rather than a brittle assumption.

*Table 7.* **Cross-Checkpoint Stability of FoG-A Rankings.** The top causal layers remain stable after the warm-up diagnostic.

| Probe Point | Speech Top-3 | Audio Top-3 | Overlap vs. Warm-up |
|---|---|---|---|
| Warm-up (5k) | $L_1, L_2, L_7$ | $L_1, L_7, L_2$ | 3/3, 3/3 |
| Mid-training | $L_1, L_2, L_8$ | $L_1, L_7, L_9$ | 2/3, 2/3 |
| Final checkpoint | $L_1, L_2, L_7$ | $L_1, L_7, L_3$ | 3/3, 2/3 |

*Table 8.* **Freeze vs. Refresh Schedule.** Refreshing $\mathcal{S}$ every epoch yields marginal gain but inflates wall-clock by 19%.

| Alignment Schedule | Speech FAD $\downarrow$ | Audio FAD $\downarrow$ | Relative Time |
|---|---|---|---|
| **Freeze @ Epoch 1 (Ours)** | **1.29** | **2.56** | $1.00\times$ |
| Refresh set every epoch | 1.28 | 2.55 | $1.19\times$ |

# B. Additional Robustness Ablations

This section consolidates ablations on hyperparameter sensitivity, timestep-adaptive routing, projection head design, internal self-alignment baselines, and a Shallow-REPA cross-architecture comparison. All numbers are obtained under Config B

unless otherwise noted.

## B.1. Static vs. Timestep-Adaptive Alignment

Figure 1(b) shows that a Dynamic Transition emerges at $t \approx 0.5$, raising the question of whether AG-REPA's static $S$ misses temporally local bottlenecks. Because the FoG-A score in Equation (9) is an expectation over $t \in [0, 1]$, any layer that is causally active during the transition accumulates enough attribution to be selected; the static Top-3 set thus acts as a "union" that covers both global and transitional bottlenecks. Empirically, a timestep-adaptive routing variant (with time-variant projection heads) yields only marginal improvement at substantially higher routing complexity (Table 9).

*Table 9.* **Static vs. Timestep-Adaptive Alignment.** Static AG-REPA matches timestep-adaptive routing while avoiding time-variant projection heads.

| Alignment Strategy | Routing Complexity | Speech FAD ↓ | Audio FAD ↓ |
|---|---|---|---|
| **Static AG-REPA (Ours)** | Low (fixed heads) | **1.29** | **2.56** |
| Timestep-Adaptive | High (time-variant heads) | 1.28 | 2.55 |

## B.2. Hyperparameter Sensitivity: $K$ and $\lambda_k$

We sweep both the number of aligned layers $K$ and the choice of weighting $\lambda_k$ (Equation (11)). Table 10 shows that $K = 3$ with FoG-A-derived weights is the optimal balance: smaller $K$ leaves critical causal layers unaligned, while larger $K$ over-regularizes the velocity field and hurts generative flexibility. Equal weighting at $K = 3$ also underperforms FoG-A weighting, confirming that the attribution-proportional weighting in Equation (11) is not a cosmetic choice.

*Table 10.* **Sensitivity to $K$ and $\lambda_k$.** $K = 3$ with FoG-A weighting is the optimal trade-off.

| Setting | Speech FAD ↓ | Audio FAD ↓ |
|---|---|---|
| $K = 2$, FoG-A $\lambda$ | 1.31 | 2.60 |
| $K = 3$, equal $\lambda$ | 1.32 | 2.59 |
| $K = 3$, **FoG-A $\lambda$ (Ours)** | **1.29** | **2.56** |
| $K = 4$, FoG-A $\lambda$ | 1.30 | 2.58 |

## B.3. Projection Head Design: MLP vs. Convolutional

iREPA (Singh et al., 2026) advocates Conv2d projection heads to preserve 2D spatial patch structure in vision. In our audio setting, representations are 1D temporal sequences and the alignment target is itself a globally-pooled teacher embedding (Equations (4) and (6)). To give convolutional alternatives their best chance, we apply 1D and 2D convolutional heads to the pre-pooled token sequence and compare them with our default 2-layer MLP on the pooled descriptor. As shown in Table 11, the pooled MLP attains the best FAD with the fewest parameters: local temporal structure inside the projection does not help match a globally-pooled teacher signal, and fine-grained acoustic detail is preserved by the DiT residual stream rather than by the projection head.

*Table 11.* **Projection Head Comparison.** The pooled MLP attains the best FAD with the fewest parameters.

| Projection Head | Applied to | Params | Speech FAD ↓ | Audio FAD ↓ |
|---|---|---|---|---|
| 2D Conv (iREPA-style) | Pre-pooled tokens | 0.29 M | 1.32 | 2.61 |
| 1D Conv ($K = 3$) | Pre-pooled tokens | 0.24 M | 1.31 | 2.58 |
| **2-layer MLP (Ours)** | Pooled descriptor | **0.18 M** | **1.29** | **2.56** |

## B.4. Comparison with Internal Self-Alignment

A complementary line of work on internal self-alignment (Haghighi et al., 2026) replaces external teachers with the model's own deeper features. This addresses an orthogonal axis to our contribution: AG-REPA is about *layer selection*, not *supervision source*. To isolate the two factors, we contrast static-heuristic alignment at L4,8,12 against AG-REPA's FoG-A selection, under both internal self-alignment and external Whisper/BEATs teachers (Table 12). Applying FoG-A selection

to internal alignment significantly improves over the static LayerSync-style baseline, confirming that the selection rule transfers across supervision sources. External Whisper/BEATs teachers remain the strongest signal, because they inject rich continuous acoustic anchors that the model cannot self-generate from sparse tokens.

*Table 12.* **Internal Self-Alignment vs. External Teachers.** Numbers are Speech FAD / Audio FAD. AG-REPA's selection rule transfers across supervision sources.

| Supervision Source | Static (L4,8,12) | + AG-REPA Selection |
|---|---|---|
| Base (no layer align.) | 1.84 / 3.45 | — |
| Internal self-alignment | 1.53 / 2.97 | 1.40 / 2.74 |
| Whisper / BEATs teachers | 1.45 / 2.88 | **1.29 / 2.56** |

## B.5. Shallow REPA vs. AG-REPA Across Architectures

A direct cross-architecture comparison between Shallow REPA (hard-coded $L_1$–$L_3$) and AG-REPA is presented in Table 5 of the main text. We summarize the key observation here for completeness: Shallow REPA is competitive on our DiT (the architecture for which the heuristic was calibrated), but its advantage erodes on other backbones because the FoG-A peak location varies across architectures and tokenizers. AG-REPA re-runs FoG-A per architecture and consistently outperforms the fixed shallow heuristic, with the gap widening on F5-TTS, Voicebox, and CosyVoice (see Table 5).

# C. Evaluation Details

**FAD Embedding Backbone.** All FAD scores reported in the main paper and appendix are computed using VGGish embeddings, with audio resampled to 16 kHz prior to feature extraction.

**Objective-Metric Aggregation.** WER and FAD are averaged over three independently initialized runs and reported as descriptive objective metrics. Because three seeds are insufficient for a reliable parametric significance test, we do not report $p$-values for WER/FAD and instead use them to compare consistent trends across alignment strategies.

**MOS Protocol.** For all MOS evaluations, 20 raters scored 40 randomly drawn samples per method under a randomized, blind protocol on a 1–5 Likert scale. Reported numbers in Tables 2 and 4 are mean±standard error. The 95% confidence intervals for AG-REPA are $[4.02, 4.22]$ (speech) and $[3.80, 4.08]$ (audio).

**Statistical Significance.** Against the strongest fixed mid-layer baseline (REPA @ L4,8,12), AG-REPA's MOS gain is statistically significant under a paired two-sided $t$-test, with $p = 0.004$ for speech and $p = 0.011$ for audio. To avoid treating repeated ratings as independent samples, the paired test is conducted on sample-level MOS after averaging across raters for each sample.

