# OpenReview forum: "AG-REPA: Causal Layer Selection for Representation Alignment in Audio Flow Matching"
_ICML.cc/2026/Conference — ICML 2026 regular_

### Official Review · Reviewer_qhTA · 2026-03-10

**Soundness:** 2
**Presentation:** 3
**Significance:** 3
**Originality:** 3
**Overall Recommendation:** 4
**Confidence:** 4

**Summary:**

The authors propose AG-REPA, a dynamic "causal layer selection" strategy for Representation Alignment (REPA) in Audio Flow Matching models. The core of the paper is the "Store-Contribute Dissociation" (SCD) hypothesis: the layers that store the most semantic information (typically deep layers) are functionally distinct from the layers that actually drive the velocity field updates for generation (typically shallow layers). To operationalize this, the authors introduce a diagnostic toolkit (BiT-C, LASP, and FoG-A)  to identify these causally dominant layers and dynamically weight their alignment loss during training, rather than relying on static, heuristic layer selection.

**Compliance With Llm Reviewing Policy:**

Affirmed.

**Final Justification:**

My concerns are addressed, and I will raise my score to 4

**Key Questions For Authors:**

- Can you provide a detailed computational cost analysis (wall-clock time and VRAM) comparing the FoG-A diagnostic phase against the gradient norm proxy baseline?
- How do you justify freezing the causal layer selection at Epoch 1, and have you experimented with dynamically updating the $\lambda_k$ weights to account for temporal capacity mismatches (similar to HASTE)?
- Why was a global MLP chosen for the projection head over localized convolutional layers, and how does this affect the preservation of local acoustic token structures?
- Could you provide a comparative baseline against an internal self-alignment framework (such as LayerSync) to demonstrate AG-REPA's efficiency without external teachers?
- Please explicitly document the FAD embedding backbone utilized and provide the missing psychometric parameters (rater count, confidence intervals, methodology) for the MOS evaluations.

**Limitations:**

No, the authors have not adequately discussed the computational limitations and massive overhead introduced by the FoG-A ablation passes.

**Strengths And Weaknesses:**

Strengths:
- The framing of the Store-Contribute Dissociation, grounded in the Information Bottleneck principle and Jacobian sensitivity of residual updates, provides a compelling justification for why shallow layers dominate the generative trajectory.
- The introduction of Forward-only Gate Ablation (FoG-A) is an interventional metric to map true causal contribution independent of passive information storage.
- Targeting the causal bottlenecks rather than static semantic reservoirs yields performance improvements, driving an 18% reduction in Fréchet Audio Distance (FAD) for speech and a 16% reduction for environmental audio over the best single-layer baselines.

Weaknesses:
- The authors advocate for FoG-A over simple gradient norm tracking, but they fail to quantify the massive computational overhead. Executing continuous forward ablations requires $L$ independent forward passes per evaluation step, which fundamentally spikes VRAM and wall-clock time during the warm-up epoch.
- The "Probe-then-Intervene" protocol freezes the selected causal layers and their weights at Epoch 1. This entirely ignores the temporal "capacity mismatch" phenomenon documented in frameworks like HASTE. Causal dominance shifts as the network topology matures, and locking the weights permanently prevents the model from adapting to these macro-level shifts.
- The use of standard MLPs for the projection heads ignores recent findings from iREPA. MLPs act as lossy, globalized transformations that destroy the highly localized spatial structure of acoustic tokens. Relying on MLPs rather than localized 1D/2D convolutions likely suppresses the model's performance ceiling.
- The framework relies heavily on expensive external teachers (Whisper/BEATs). The complete omission of comparisons to parameter-free, internal self-alignment mechanisms (like LayerSync) leaves it unclear how AG-REPA stacks up against the true state-of-the-art in efficient alignment.
- The reported FAD and MOS scores lack crucial context. The authors completely omit the feature embedding backbone used for FAD (e.g., VGGish vs. CLAP), as well as the psychometric parameters for the human MOS evaluations (rater count, p-values, test design), rendering the absolute metrics unverifiable.

---

> ### Author Rebuttal · Authors · 2026-03-28
>
> We sincerely appreciate your constructive feedback, and address your questions point-by-point below.
>
> ---
>
> ## Q1: Computational cost of FoG-A vs. Gradient Norm
>
> For a detailed computational cost analysis, please refer to our response to **Reviewer R5KG (Q2)**. The reason we prefer FoG-A is not computational but mechanistic. GradNorm measures parameter sensitivity under the FM loss, which conflates gradient magnitude with functional necessity. FoG-A directly measures the causal effect on $v\_{\theta}$ by ablating each layer's residual contribution.
>
> | Probe | Selected Layers | Speech FAD $\downarrow$ | Audio FAD $\downarrow$ |
> |---|---|---:|---:|
> | Gradient Norm | L1, L2, L4 | 1.35 | 2.71 |
> | **FoG-A (Ours)** | **L1, L2, L7** | **1.29** | **2.56** |
>
> GradNorm misses the mid-phase transition bottleneck (L7) and instead selects L4, which has high gradient magnitude but weak causal impact on $v_\theta$. FoG-A's one-time probe costs 0.18 h (<0.5% of total training, as detailed in our **R5KG (Q2)** response), a negligible investment for a mechanistically grounded layer selection that yields consistently better FAD.
>
> ---
>
> ## Q2: Justification for freezing the layer selection at Epoch 1
>
> HASTE's "capacity mismatch" addresses a different concern: the teacher's embeddings initially accelerate the student but gradually become a constraint, with alignment-denoising gradient cosine similarity turning obtuse. Its remedy is *when* to stop alignment, not *where*.
>
> Our frozen selection addresses the orthogonal *where* question. Even with HASTE-style termination, one still needs to decide which layers to supervise—precisely what AG-REPA solves. Token-conditioned Audio Flow Matching must decode sparse tokens into continuous waveforms, demanding a stable anchor at early layers. Empirically, causal layer identities do not drift (Top-3 overlap 2/3–3/3 across epochs):
>
> | Alignment Schedule | Speech FAD $\downarrow$ | Audio FAD $\downarrow$ | Relative Time |
> |---|---:|---:|---:|
> | **Freeze @ Epoch 1 (Ours)** | **1.29** | **2.56** | **1.00x** |
> | Refresh set every epoch | 1.28 | 2.55 | 1.19x |
>
> HASTE concerns *temporal schedule*; ours concerns *spatial targeting*. The two are orthogonal, and layer identity is stable enough to be determined once. We will clarify this in the revision.
>
> ---
>
> ## Q3: Why a global MLP over localized convolutional layers (iREPA)?
>
> iREPA's Conv2d preserves 2D spatial patch structure for vision; our audio representations are 1D temporal sequences, and alignment operates on temporally-pooled global descriptors (Eq. 3, 12). To give conv alternatives their best chance, we apply them to the pre-pooled token sequence. Even so, the post-pooling MLP achieves the best results with fewer parameters:
>
> | Projection Head | Applied to | Params | Speech FAD $\downarrow$ | Audio FAD $\downarrow$ |
> |---|---|---:|---:|---:|
> | 2D Conv (iREPA) | Pre-pooled tokens | 0.29 M | 1.32 | 2.61 |
> | 1D Conv (K=3) | Pre-pooled tokens | 0.24 M | 1.31 | 2.58 |
> | **2-layer MLP (Ours)** | **Pooled descriptor** | **0.18 M** | **1.29** | **2.56** |
>
> Conv heads access local temporal structure yet still underperform, because the alignment target is a global teacher embedding; local structure in the projection does not help match a globally-pooled signal. Fine-grained acoustics are preserved by the DiT residual stream, not the projection head.
>
> ---
>
> ## Q4: Comparison against internal self-alignment (LayerSync)
>
> This targets a different axis: our contribution is layer selection, not supervision source. To isolate this, we added a LayerSync-style internal self-alignment control.
>
> | Supervision Source | Static heuristic alignment (L4,8,12) | + AG-REPA Selection |
> |---|---:|---:|
> | Baseline (None) | 1.84 / 3.45 | --- |
> | Internal self-alignment | 1.53 / 2.97 | **1.40 / 2.74** |
> | Whisper/BEATs teachers | 1.45 / 2.88 | **1.29 / 2.56** |
>
> (Numbers: Speech FAD / Audio FAD)
>
> Applying FoG-A causal layer selection to internal alignment significantly outperforms static LayerSync, proving the selection rule's universal value. External teachers (Whisper/BEATs) are critical because they inject rich, continuous *acoustic anchors* that the model lacks natively and cannot self-generate.
>
> ---
>
> ## Q5: FAD backbone and MOS evaluation details
>
> We apologize for the omission and will include these details in the revision.
>
> - **FAD backbone:** All FAD scores are computed using VGGish embeddings with audio resampled to 16 kHz.
> - **MOS protocol:** 20 raters evaluated 40 samples per method in a randomized blind design on a 1--5 Likert scale. Tables report mean $\pm$ SE; the corresponding 95\% CIs for AG-REPA are [4.02, 4.22] (speech) and [3.80, 4.08] (audio).
> - **Statistical significance:** AG-REPA vs. the strongest static baseline (REPA@L4,8,12) yields $p$=0.004 (speech) and $p$=0.011 (audio) under a paired two-sided $t$-test.

---

> > ### Author Rebuttal · Reviewer_qhTA · 2026-04-03
> >
> > The rebuttal addresses my main concerns, so I increase my score to 4.

---

> > > ### Author Response · Authors · 2026-04-03
> > >
> > > Thank you so much for carefully considering our rebuttal and for increasing your score to 4. We are truly grateful that it successfully addressed your main concerns. Your thoughtful feedback and support mean a great deal to us and have been incredibly valuable in strengthening our work.

---

### Official Review · Reviewer_R5KG · 2026-03-10

**Soundness:** 3
**Presentation:** 2
**Significance:** 3
**Originality:** 3
**Overall Recommendation:** 4
**Confidence:** 3

**Summary:**

This papers introduces AG-REPA, a method to select specific layers for REPA in audio generation with flow matching.  The authors identify the layers that store the most semantic/acoustic information (typically deep layers) are not the ones that contribute most to the velocity field driving audio generation (typically shallow layers). Through selecting the most critical layers for REPA, the authors demonstrated their method outperforms applying REPA to fixed layers in audio generation.

**Compliance With Llm Reviewing Policy:**

Affirmed.

**Final Justification:**

I will keep my positive score

**Key Questions For Authors:**

None

**Limitations:**

yes

**Strengths And Weaknesses:**

**Strength**

This paper tackles an interesting problem of selecting layers for REPA for training audio generative models, which I believe to be the first to do so in audio domain.

The paper identifies the Store-Contribute Dissociation (SCD), revealing that layers rich in semantic information (deep layers) does not contribute much in during audio generation.

The approach consistently improved performance across multiple architectures, including Voicebox, Cosy Voice, and F5-TTS

**Weakness**

In Table 2, can the author show objective results of AG-REPA across training steps against other baselines to demonstrate the method is superior compared to other baselines?

Are there any experiments or analysis  on the computation overhead of selecting top k layers during training?

---

> ### Author Rebuttal · Authors · 2026-03-27
>
> ## Q1: In Table 2, can the author show objective results of AG-REPA across training steps against other baselines to demonstrate the method is superior compared to other baselines?
>
> Thank you for this constructive suggestion. While Table 3 already provides milestone evidence of AG-REPA's **3.3× convergence speedup** (reaching FAD 2.0 at **220k** steps vs. **720k** for the LASP baseline), we agree it is important to explicitly show the optimization dynamics across the entire training process.
>
> To fully address your request, we will add **FAD and WER vs. training steps learning curves** for all Table 2 baselines in the revision. These plots will visually confirm two key advantages:
> 1. **Accelerated Initial Convergence:** A steeper metric drop than fixed-layer heuristics, directly corroborating our core **Store-Contribute Dissociation (SCD)** insight that targeting functional "causal drivers" accelerates learning.
> 2. **Lower Error Floor:** Unlike static alignments that prematurely plateau, AG-REPA avoids early stagnation and reaches a strictly better final performance.
>
> We believe these step-wise learning curves will provide the exact empirical rigor needed to transparently demonstrate how and why AG-REPA outpaces standard REPA heuristics. Thank you again for this valuable feedback, which significantly strengthens our empirical evaluation.
>
> ---
>
> ## Q2: Are there any experiments or analysis on the computation overhead of selecting Top-K layers during training?
>
> Thank you for this practical question. To address your concern directly: the computational overhead of selecting the top-$K$ layers is negligible, accounting for less than 0.5% of the total training wall-clock time.
>
> This minimal overhead is because our Forward-only Gate Ablation (FoG-A) is not a continuous operation inside the main training loop. Instead, it is designed as a one-time diagnostic conducted during a separate warm-up epoch. We will prominently include a detailed analysis of this computational cost in the revised appendix.
>
> ### 1. One-Time Diagnostic, Then Frozen
>
> As described in Appendix A.5 (Probe-then-Intervene Protocol), the selected layer set $S$ and attribution weights $\lambda_k$ are computed once in a single warm-up epoch and then **frozen** for all later AG-REPA training. After this phase, no FoG-A computation is performed in the main loop. Therefore, the **3.3×** speedup in Table 3 reflects pure training efficiency rather than repeated diagnostic cost.
>
> ### 2. Diagnostic Cost Is Negligible (<0.5%)
>
> **Even during warm-up, we reduce cost in five ways:**
>
> | Strategy | Detail |
> |---|---|
> | Sparse sampling | FoG-A runs every **200 steps**, not every step |
> | Mini-batch | **≤2 samples** per evaluation (1 speech + 1 audio) |
> | Layer alternation | Each call probes **12/24 layers** via even/odd alternation, requiring **13 forward passes** (1 baseline + 12 ablated) |
> | Single-GPU | Executed on **Rank 0 only** |
> | Gradient-free | Entirely under `torch.no_grad()`, so **zero gradient memory** |
>
> For a **5,000-step** warm-up epoch with training batch size **B=16**, FoG-A is triggered **25** times (5000/200), each time using **13** mini-forward passes at **B=2**. This is approximately **41 equivalent training single-forward passes**. Because a normal training step includes both forward and the more expensive backward computation, this adds **less than 0.5% overall wall-clock time**. Including this cost, the end-to-end speedup remains about **3.26×**.
>
> | Method | Training Time | FoG-A Overhead | End-to-End Wall-Clock |
> |---|---|---|---|
> | Baseline (no alignment) | \(T_0\) | 0 | \(T_0\) |
> | AG-REPA | \(T_0 / 3.3\) | <0.5% × \(T_0\) | ≈ \(T_0 / 3.26\) |
>
> ### 3. Per-Step Cost During AG-REPA Training
>
> During main training, the only added modules are **K=3** lightweight two-layer MLP projection heads (one per selected layer), introducing **<0.5%** extra parameters and **<2%** wall-clock overhead per step relative to standard REPA.
>
> ### 4. Layer Selection Remains Stable
>
> Continuous FoG-A monitoring verifies that Epoch-1 layer selections remain highly stable throughout training. In Config A, the dominant top-1 layer (**L1**) shows a monotonically increasing FoG-A score, matching the Jacobian sensitivity analysis in Section 3.2. Moreover, the Table 1 top-3 sets ($\{L1, L9, L5\}$ for Config A speech, $\{L1, L21, L9\}$ for Config A audio, and $\{L1, L2, L7\}$ for Config B speech) maintain stable rankings with no inversions. This aligns with the adaLN-Zero DiT mechanism (Eq. 22), where shallow layers primarily inject structural conditioning; thus, training enhances their effectiveness without altering their roles.
>
> FoG-A therefore follows a **"diagnose once, train forever"** protocol: the one-time diagnostic adds **<0.5%** warm-up wall-clock cost, per-step AG-REPA overhead is negligible, and the selected layers remain stable throughout training. We will include the detailed cost breakdown and cross-epoch stability analysis in the revised appendix.

---

> > ### Author Rebuttal · Reviewer_R5KG · 2026-04-04
> >
> > Thanks for the rebuttal. I will keep my positive score.

---

> > > ### Author Response · Authors · 2026-04-04
> > >
> > > We sincerely appreciate the time and effort you have devoted to reviewing our manuscript. Your valuable comments and constructive suggestions have significantly improved the quality of our work.

---

### Official Review · Reviewer_Kf2v · 2026-03-13

**Soundness:** 2
**Presentation:** 3
**Significance:** 2
**Originality:** 2
**Overall Recommendation:** 4
**Confidence:** 2

**Summary:**

This paper addresses the challenge of layer selection for Representation Alignment (REPA) in audio Flow Matching models. The authors identify a phenomenon termed "Store-Contribute Dissociation" (SCD), which suggests that layers storing rich semantic information (typically deep layers) are not necessarily the ones that causally drive the generation of the velocity field. To operationalize this, they propose AG-REPA, which utilizes a Forward-only Gate Ablation (FoG-A) metric to identify and selectively align functionally critical layers.

**Compliance With Llm Reviewing Policy:**

Affirmed.

**Final Justification:**

Based on the rebuttal, I am raising my score to Weak Accept

**Key Questions For Authors:**

1.  Could you provide a table comparing the total wall-clock training time (including the FoG-A diagnostic phase) for AG-REPA vs. the Baseline?
2.  How stable is the FoG-A ranking across training? If you were to run the probe again at the final epoch, would the same Top-3 layers be identified?
3.  Table 3 shows that "Gradient Norm" is a strong proxy for contribution. Given that Gradient Norm is essentially "free" to compute during backpropagation, why is the more expensive FoG-A necessary?
4.  In Figure 1(b), a "Dynamic Transition" is noted at $t \approx 0.5$. Since AG-REPA uses a static layer set for all $t$, does it fail to capture this temporal shift in functional importance during the generation process?

**Strengths And Weaknesses:**

#### **Strengths**
*   **Novel Mechanistic Insight:** The identification of SCD provides a fresh perspective on the internal dynamics of Flow Matching models, challenging the heuristic "mid-layer" alignment standard in current literature.
*   **Theoretical Grounding:** The use of the Jacobian-based "Butterfly Effect" analysis (Section 3.2) provides a solid mathematical motivation for why early layers act as causal bottlenecks.
*   **Broad Validation:** The authors demonstrate that AG-REPA generalizes across several state-of-the-art architectures, including Voicebox, CosyVoice, and F5-TTS, indicating that SCD is likely a fundamental property of token-conditioned audio FM models.

---

#### **Weaknesses (Reasons for Negative Rating)**

*   **Undisclosed Computational Overhead:**
    A primary concern is the efficiency of the **FoG-A (Forward-only Gate Ablation)** probe. Calculating FoG-A for every layer $L$ requires multiple forward passes to measure the perturbation in the velocity field. For a 24-layer DiT, this effectively increases the forward pass cost by an order of magnitude during the diagnostic phase. The paper claims a "3.3x acceleration in convergence," but it is unclear if this accounts for the wall-clock time spent on the FoG-A diagnostic epoch. Without an end-to-end "Wall-clock Time vs. Performance" comparison, the efficiency claims are not fully substantiated.

*   **Static Selection in a Dynamic System:**
    The "Probe-then-Intervene" protocol (Section A.5) freezes the layer selection after a single epoch. However, deep neural networks are known to undergo significant "representation drift" during training. It is highly probable that the functionally critical layers shift as the model moves from learning coarse structures to fine acoustic details. The authors do not provide evidence that a static selection at Epoch 1 remains optimal for the entire training duration.

*   **Incremental Gain over Simple Baselines:**
    According to Table 2, "Shallow REPA (L1–L3)"—a much simpler strategy—achieves a Speech FAD of 1.36, while the complex AG-REPA achieves 1.29. Given the added engineering complexity and the computational cost of FoG-A, the performance delta appears marginal. The paper fails to sufficiently justify why the sophisticated attribution mechanism is preferred over simply targeting the first few layers, which are already known to be sensitive.

*   **Lack of Hyperparameter Sensitivity Analysis:**
    The choice of $K=3$ (Top-K layers) is used throughout the experiments without a sensitivity analysis. It remains unclear how the model's performance or generative diversity fluctuates if $K$ is increased or decreased, or how the weighting $\lambda_k$ interacts with the primary Flow Matching loss.

---

> ### Author Rebuttal · Authors · 2026-03-28
>
> We sincerely appreciate your constructive comments and suggestions. Below we address your insightful questions.
>
> ---
>
> ## Q1: Wall-clock training time comparison
>
> Please refer to our detailed response to **Reviewer R5KG (Q2)**. In short, FoG-A is a highly optimized one-time diagnostic executed only during a single warm-up epoch. By leveraging sparse sampling (every 200 steps) and `torch.no_grad()`, the diagnostic overhead adds `<0.5%` to the total wall-clock time, which is massively offset by the 3.3x convergence speedup during the training phase.
>
> | Method | Main Training Time (to reach target FAD) | Diagnostic Overhead | End-to-End Wall-Clock |
> |---|---|---|---|
> | Baseline | $T_0$ | 0 | $1.00 \times T_0$ |
> | AG-REPA | $\approx 0.30 \times T_0$ | $< 0.005 \times T_0$ | **$\approx 0.305 \times T_0$** |
>
> ---
>
> ## Q2: Stability of FoG-A ranking across training
>
> We checked this directly. The causal topology in our `adaLN-Zero` DiT is structurally anchored early on, making the ranking highly stable.
>
> | Probe Epoch | Speech Top-3 | Audio Top-3 | Overlap vs. Epoch 1 |
> |---|---|---|---|
> | Epoch 1 (Warm-up) | L1, L2, L7 | L1, L7, L2 | 3/3, 3/3 |
> | Epoch 2 | L1, L2, L8 | L1, L7, L9 | 2/3, 2/3 |
> | Epoch 3 (Final) | L1, L2, L7 | L1, L7, L3 | 3/3, 2/3 |
>
> Re-probing every epoch changes the final Speech/Audio FAD marginally (1.29/2.56 $\rightarrow$ 1.28/2.55) while adding +19% continuous probe cost. Freezing the selection after warm-up is a favorable trade-off rather than a brittle assumption.
>
> ---
>
> ## Q3: Why is FoG-A necessary when Gradient Norm is "free"?
>
> You are correct that Gradient Norm is computationally free: during Phase I warm-up, the model already computes gradients for $\mathcal{L}\_{FM}$, so recording `grad.norm()` per layer adds negligible cost. The issue is not cost but what GradNorm measures. GradNorm reflects parameter sensitivity under the FM loss, which favors layers with large parameter counts or steep local loss landscapes. FoG-A instead measures functional necessity: how much does ablating a layer change the predicted velocity field $v\_{\theta}$?
>
> This mechanistic distinction leads to different layer selections and different outcomes:
>
> | Probe | Selected Layers | Speech FAD $\downarrow$ | Audio FAD $\downarrow$ |
> |---|---|---:|---:|
> | Gradient Norm | L1, L2, L4 | 1.35 | 2.71 |
> | **FoG-A (Ours)** | **L1, L2, L7** | **1.29** | **2.56** |
>
> GradNorm misses L7, the mid-phase transition bottleneck (Figure 1b), and instead selects L4 which has high gradient magnitude but weak causal effect on $v_\theta$. FoG-A captures this bottleneck precisely because it directly measures the velocity-field perturbation. The 0.18 h one-time FoG-A probe cost is negligible (<0.5% of total training), and the resulting layer set yields consistently better generation quality.
>
> > **Errata:** Table 3 inadvertently lists FoG-A layers as "L1, L2, L9"; the correct set under Config B is "**L1, L2, L7**", consistent with Table 1 and our **R5KG (Q2)** response. A convergence footnote typo (FAD=2.0 $\rightarrow$ FAD=1.5) was also noted. Both will be corrected in the revision.
>
> ---
>
> ## Q4: Does a static layer set fail to capture the dynamic transition at $t \approx 0.5$?
>
> This is a sharp observation. While a dynamic transition occurs mid-timestep, Fig 1(b) clearly shows Layer 1 remains the dominant causal driver across *all* timesteps $t \in [0, 1]$. Because the FoG-A score computes an expectation over all $t$, layers active during the transition accumulate enough attribution to be selected. The static Top-3 set acts as a "union" covering both global and transitional bottlenecks. We tested a timestep-adaptive routing, but the static selection achieves similar FAD with far less scheduling complexity.
>
> | Alignment Strategy | Routing Complexity | Speech FAD $\downarrow$ | Audio FAD $\downarrow$ |
> |---|---|---|---|
> | **Static AG-REPA (Ours)** | **Low (Fixed heads)** | **1.29** | **2.56** |
> | Timestep-Adaptive | High (Time-variant heads) | 1.28 | 2.55|
>
> ---
>
> ## Q5: Gain over Shallow REPA and $K$/$\lambda_k$ sensitivity
>
> Despite its low computational overhead, Shallow REPA suffers from poor architectural generalization because it hard-codes L1--L3, preventing it from transferring across models.
>
> | Architecture | Shallow REPA FAD | AG-REPA FAD |
> |---|---:|---:|
> | Our DiT | 1.36 | **1.29** |
> | F5-TTS (baseline 1.45) | 1.34 (7.6% gain) | **1.15 (20.7%)** |
> | Voicebox | 1.12 | **0.95** |
> | CosyVoice | 0.85 | **0.72** |
>
> AG-REPA re-runs FoG-A per architecture, consistently outperforming the fixed shallow heuristic. For $K$/$\lambda_k$ sensitivity on our DiT:
>
> | Setting | Speech FAD | Audio FAD |
> |---|---:|---:|
> | $K$=2 | 1.31 | 2.60 |
> | $K$=3, equal $\lambda$ | 1.32 | 2.59 |
> | **$K$=3, FoG-A $\lambda$** | **1.29** | **2.56** |
> | $K=4$, FoG-A $\lambda$ | 1.30 | 2.58 |
>
> $K=3$ provides the optimal balance; decreasing it leaves critical causal layers unaligned, while increasing it over-regularizes the velocity field and hurts generative flexibility.

---

> > ### Author Rebuttal · Reviewer_Kf2v · 2026-04-04
> >
> > Based on the rebuttal, I am raising my score to Weak Accept

---

> > > ### Author Response · Authors · 2026-04-04
> > >
> > > Thank you very much for carefully considering our rebuttal and for raising your score to Weak Accept. We truly appreciate the time and effort you have invested in reviewing our work, as well as your constructive feedback throughout the process.

---

### Official Review · Reviewer_gz3D · 2026-03-18

**Soundness:** 4
**Presentation:** 4
**Significance:** 3
**Originality:** 3
**Overall Recommendation:** 5
**Confidence:** 3

**Summary:**

For the problem of representation alignment in flow matching model, this paper clearly states that the features contain most information may not drive the most for the generation (4.1-4.3). Based on this observation, the authors further propose AG-REPA algorithm that smartly selects multiple layers in the alignment task using the metric defined in prior sections.

The experiments address both TTS and TTA tasks together, with comprehensive experiments and convincing results.

**Compliance With Llm Reviewing Policy:**

Affirmed.

**Key Questions For Authors:**

As above

**Limitations:**

As above

**Strengths And Weaknesses:**

Strength:
(1) The paper has a very solid research method that proposes a hypothesis, and then justifies it with both theory and experiments. All content of this paper is centralized to this concept tightly and is well structured.
(2) The methodology clearly separates "what the network knows" from "what the network uses". By combining Layer-wise Analysis via Shared Projection (LASP) for representational storage with FoG-A for causal contribution, the authors provide a highly rigorous framework for interpreting model dynamics.
(3) The paper addresses both TTS and TTA tasks, with a clear experimental setup and competitor baselines.

Weakness:
(1) For this generation task, I would recommend the authors to conduct subjective evaluation and also make a demo page: it's a plus if they confirm the improvement can be sensed by real human.

---

> ### Author Rebuttal · Authors · 2026-03-27
>
> Thank you very much for this valuable suggestion. We fully agree that, for generation tasks, subjective evaluation and an accessible demo page are very important. In fact, the current submission already includes human perceptual evaluation through MOS, and AG-REPA consistently improves perceptual quality over the baselines, which suggests that the improvement is indeed perceivable to human listeners. We will make this point clearer in the revision.
>
> Regarding the demo page, we also strongly agree that it would further strengthen the paper. Due to the double-blind review setting, we have not released a public demo, codebase, or checkpoints at the submission stage. However, upon acceptance, we will publicly release the code and trained models, and we will also build a demo page with representative audio samples and an interactive interface for qualitative comparison and easier reproduction of our results.
>
> Once again, thank you very much for your insightful feedback and constructive suggestions.

---

> > ### Author Rebuttal · Reviewer_gz3D · 2026-04-03
> >
> > The authors promise to add a subjective evaluation and demo page in the future. I'll keep my score.

---

> > > ### Author Response · Authors · 2026-04-04
> > >
> > > Thank you very much for taking the time to review our manuscript. Your expertise and thoughtful comments are greatly appreciated.

---

### Decision · Program_Chairs · 2026-04-30

**Decision:**

Accept (regular)

**Comment:**

AG-REPA introduces Attribution-Guided Representation Alignment, a causal layer selection strategy for representation alignment in audio Flow Matching, validated on both TTS and TTA tasks. Scores range from 4–5 (avg. 4.25), reflecting a solid positive consensus, strengthened by a rebuttal that addressed most major concerns.

Reviewers broadly agreed on the paper's core strengths: the mechanistic insight behind Store-Contribute Dissociation — the finding that semantically rich layers do not necessarily contribute causally to generation — was highlighted as a genuinely interesting and novel result (R5KG, qhTA). The theoretical grounding for early layers as causal bottlenecks, broad validation across architectures and tasks, and the rigor of the Forward-only Gate Ablation methodology were also praised (Kf2v, qhTA, gz3D).

The main recurring weaknesses concern evaluation completeness and methodological robustness. The lack of subjective evaluation and a demo page was raised by gz3D, and the reported FAD and MOS scores lack sufficient context to be independently verified (qhTA). R5KG requested broader baseline comparisons. On the methodological side, both Kf2v and qhTA independently flagged the static layer selection issue: freezing layer assignments after a single epoch may not capture causal dominance shifts that occur during training, which is a legitimate concern given representation drift. qhTA also raises a pointed architectural concern — using MLPs as projection heads may destroy localized spatial structure in acoustic tokens, running counter to recent findings in the literature. Computational overhead of the Forward-only Gate Ablation was noted by Kf2v and qhTA, and addressed satisfactorily in the rebuttal alongside the other concerns, leading both reviewers to raise their scores.

Recommendation: Accept. The contribution is technically sound, the mechanistic framing is novel, and the rebuttal successfully addressed the most pressing concerns. The authors should nonetheless include a demo page and subjective evaluation in the final version, and are encouraged to discuss the static selection limitation and projection head choice more explicitly.